# R³L: Reasoning 3D Layouts from Relative Spatial Relations

Zhifeng Gu [* 1]   Yuqi Wang [* 1]   Bing Wang [1]

## Abstract

Relative spatial relations provide a compact representation of spatial structure and are fundamental to relative spatial reasoning in 3D layout generation. Recent works leverage Multimodal Large Language Models (MLLMs) to infer such relations, but the inferred relations are often unreliable and are typically handled with post-hoc heuristics. In this paper, we propose **R³L**, a general framework that improves the reliability and consistency of relative spatial reasoning for 3D layout generation. Our key motivation is that multi-hop reasoning requires repeated reference-frame transformations, which accumulate errors in inferred relations and lead to semantic and metric drift. To mitigate this, we propose invariant spatial decomposition to break coupled relation chains, and consistent spatial imagination to promote self-consistency through an imagine-and-revise loop. We further introduce supportive spatial optimization to ease pose optimization via global-to-local coordinate reparameterization. Extensive experiments across diverse scene types and instructions demonstrate that **R³L** produces more physically feasible and semantically consistent layouts. Notably, our analysis shows that resolving frame-induced inconsistencies is crucial for reliable multi-hop relative spatial reasoning. The code is available at https://github.com/Neal2020GitHub/R3L.

## 1. Introduction

Relative spatial relations describe object-to-object spatial relationships under a specified reference frame (Levinson, 2003), providing a compact and human-aligned representation of spatial structure. They are essential for vision and

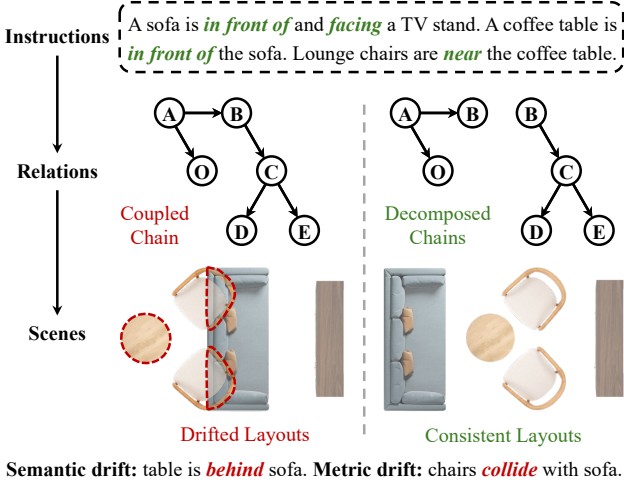

*Figure 1.* Previous methods (left) reason over coupled relation chains, where repeated reference-frame transformations accumulate errors in inferred relations and lead to drifted layouts. In contrast, **R³L** (right) decomposes long coupled chains into shorter sub-chains to reduce reference-frame transformations and error accumulation, producing feasible and consistent layouts.

robotics tasks that require grounding high-level relational intent into concrete spatial arrangements (Gervet et al., 2023; Shao et al., 2025; Grauman et al., 2022). 3D layout generation, a representative of these tasks, formulates scene structure through relative spatial relations and instantiates them into a final layout. At the core of inferring relative spatial relations is relative spatial reasoning (Gardner, 1983; Li & Gleitman, 2002; Lee et al., 2025), the ability to reason about relations across multiple objects.

In conventional 3D layout generation, layouts are largely generated from hand-crafted rules or dataset-specific priors (Merrell et al., 2011; Paschalidou et al., 2021; Tang et al., 2024), which often generalize poorly. Recent works instead leverage Multimodal Large Language Models (MLLMs) (Hurst et al., 2024; Comanici et al., 2025) to reason over relative spatial relations, and instantiate the inferred relations into object poses (Yang et al., 2024b; Sun et al., 2025). However, prior works rarely explore the reliability of the relative spatial reasoning process, which can produce relations that are semantically inconsistent or physically infeasible.

---
[*]Equal contribution [1]Spatial Intelligence Group, The Hong Kong Polytechnic University. Correspondence to: Bing Wang <bingwang@polyu.edu.hk>.

*Proceedings of the 43ʳᵈ International Conference on Machine Learning*, Seoul, South Korea. PMLR 306, 2026. Copyright 2026 by the author(s).

Consequently, existing pipelines rely on post-hoc heuristics to make these relations "*solvable*" for downstream instantiation, such as grid-map discretization (Yang et al., 2024b; Çelen et al., 2024) or pruning conflicting relations (Sun et al., 2025), often at the cost of semantic fidelity.

A more principled alternative is to resolve such issues during relation reasoning, making the relations "*correct*" both semantically and physically. This raises a core problem: how to fully realize MLLMs' potential for relative spatial reasoning to produce semantically consistent and physically feasible relations for 3D layout generation.

In relative spatial reasoning, the main challenge often lies in multi-hop reasoning over **chains of relations**. When two objects are not directly linked, their relation must be inferred by composing per-hop (*i.e.*, pairwise) relations through intermediate objects into a relation chain. Since each hop is expressed in an object-centric frame, reasoning over such relation chains inevitably requires repeated transformations across object-centric reference frames. This forces MLLMs to repeatedly re-express intermediate conclusions under new frames, allowing errors to accumulate along the chain, as illustrated in Figure 1. Such reference-frame transformation is analogous to *mental rotation* in cognitive science (Shepard & Metzler, 1971; Zacks, 2008).

These accumulated errors manifest in two common failure modes. **Semantic drift** occurs when directional relations are re-interpreted under different reference frames. Even slight frame-induced mismatches (*e.g.*, a local-axis swap that remaps left/right to up/down) can distort directional semantics and introduce inconsistencies into the inferred relations. **Metric drift** arises when metric displacements are composed as per-hop vectors under changing reference frames. Even small geometric misalignments can compound in the inferred relations, leading to inconsistent spacing, collisions, and physically infeasible layouts.

To address these issues, we propose a general framework, **R³L**, to improve the reliability and consistency of relative spatial reasoning for 3D layout generation. We present **invariant spatial decomposition**, which partitions a scene into frame-invariant units whose intra-unit configurations are preserved under reference-frame transformations. By factorizing relations into intra- and inter-unit relations, this decomposition breaks long relation chains, thereby reducing reference-frame transformations during multi-hop reasoning and mitigating semantic drift. We further propose **consistent spatial imagination** for self-consistent relation inference. Using frame-invariant units, the MLLM imagines object placements in unit-local and global frames, detects geometric violations, and iteratively revises inconsistent relations. This improves multi-hop metric composition and mitigates metric drift, while enabling unit-level revision to correct local errors before they propagate globally. Finally, we design **supportive spatial optimization**, where global-to-local pose re-parameterization supports pose optimization. We optimize the full scene under a unified objective, updating object poses in unit-local coordinates to stabilize optimization while preserving global coherence.

Overall, our contributions are as follows:

- We introduce **R³L**, a general framework for 3D layout generation that improves the reliability and consistency of multi-hop relative spatial reasoning.
- We present invariant spatial decomposition, which partitions scenes into frame-invariant units, reducing reference-frame transformations during multi-hop reasoning.
- We propose consistent spatial imagination, a local-to-global imagine-and-revise mechanism that promotes self-consistency and feasibility during relation inference.
- We design supportive spatial optimization, which uses global-to-local pose re-parameterization to stabilize optimization and maintain global layout coherence.

Extensive experiments across diverse scene types and instructions demonstrate that **R³L** produces physically feasible and semantically consistent layouts. Ablation studies further show that the proposed modules reduce semantic and metric drift in multi-hop relation inference.

## 2. Related Work

### 2.1. Spatial Reasoning in MLLMs

Spatial reasoning, the ability to perceive and manipulate spatial relationships in the 3D world, is fundamental for embodied agents (Driess et al., 2023; Kim et al., 2024) and world models (Liu et al., 2024; Zhen et al., 2024). Powered by modern LLMs (Achiam et al., 2023; Bai et al., 2023; Guo et al., 2025; Team et al., 2023; Touvron et al., 2023a;b) and strong vision encoders (Radford et al., 2021; Caron et al., 2021; Oquab et al., 2023), MLLMs have achieved impressive visual understanding (Hurst et al., 2024; Liu et al., 2023; Bai et al., 2025; Comanici et al., 2025) but still struggle with spatial reasoning (Yang et al., 2025a; Jia et al., 2026; Wang et al., 2025a; 2026; Yang et al., 2026). To bridge this gap, recent works construct spatial datasets and finetune MLLMs to improve spatial understanding and reasoning (Chen et al., 2024; Cheng et al., 2024; Ma et al., 2024; Cai et al., 2025; Qi et al., 2025). In parallel, another line of research elicits more explicit intermediate representations, such as visualization-of-thought (Li et al., 2025), textual cognitive maps (Ouyang et al., 2025), explicit 3D representations (Ma et al., 2025), and visual drawing (Wu et al., 2025). In this work, we propose a framework to realize MLLMs' potential for relative spatial reasoning to infer semantically consistent and physically feasible relative spatial relations for 3D layout generation.

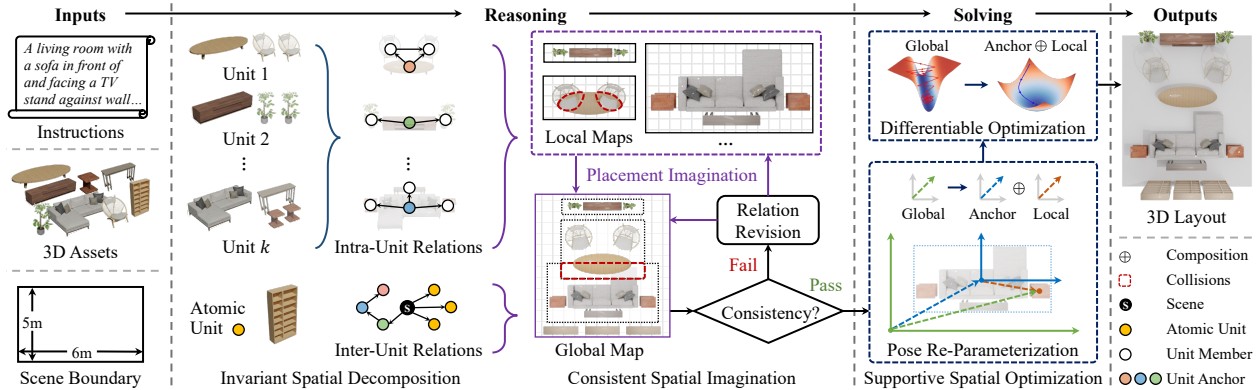

*Figure 2.* **Overview of the R³L Framework.** Given a language instruction and a set of 3D assets, **invariant spatial decomposition** partitions assets into frame-invariant units and generates intra-unit and inter-unit relations. Next, **consistent spatial imagination** runs an imagine-and-revise loop to promote self-consistency of the inferred relations. Finally, **supportive spatial optimization** applies global-to-local pose re-parameterization to stabilize optimization and outputs the final 3D layout.

## 2.2. Direct 3D Layout Generation

Recent advances in MLLMs have made open-vocabulary 3D layout generation practical. A straightforward paradigm is prompting MLLMs to directly output numerical object positions and orientations. LayoutGPT (Feng et al., 2023) leverages in-context learning by retrieving similar layouts from databases and formatting them as CSS-style demonstrations to compose new 3D layouts, while often producing physically invalid layouts. To improve layout quality, subsequent works train or align MLLMs for direct layout generation and editing through supervised fine-tuning (Bucher & Armeni, 2025; Yang et al., 2024a) and preference alignment (Yang et al., 2025c; Ran et al., 2025). Despite these advances, such approaches are typically built on curated 3D layout corpora such as 3D-FRONT (Fu et al., 2021), which limits generalization to novel scene types and open-vocabulary instructions. More recent works use agentic or tool-augmented frameworks that iteratively refine layouts using semantic and physical feedback (Berdoz et al., 2026; Yang et al., 2025b; Xia et al., 2026; Pfaff et al., 2026). However, these frameworks often rely on an explicit trial-and-error: they begin from physically infeasible drafts and repeatedly invoke external feedback tools for repair, which can be iteration-heavy and unstable. In contrast, we integrate an implicit feedback mechanism into the MLLM's reasoning process, enabling self-revision during relation inference so that local inconsistencies are corrected before they propagate globally and the resulting relations are more reliable and consistent in a single pass.

## 2.3. Constraint-driven 3D Layout Optimization

Another paradigm first infers relative spatial relations and then instantiates object poses to satisfy them. HOLODECK (Yang et al., 2024b) pioneers this direction by generating spatial relations with MLLMs and solving them with a DFS-based solver. Subsequent works extend this pipeline with multi-agent architectures (Çelen et al., 2024), richer constraint libraries (Raistrick et al., 2024; Tam et al., 2025; Aguina-Kang et al., 2024), multimodal conditioning (Zhou et al., 2025), global-local tree search (Deng et al., 2025), and hierarchical decomposition (Wang et al., 2025b; Pun et al., 2026; El Amine Boudjoghra et al., 2025). However, grid-based DFS solvers restrict the solution space to discrete candidates, which simplifies feasibility checking but often sacrifices semantic fidelity, resulting in physically valid but semantically inconsistent layouts. LayoutVLM (Sun et al., 2025) addresses this by introducing a layout representation that combines numerical pose estimates with spatial relations and refining it via differentiable optimization to better balance semantic coherence and physical plausibility. Nevertheless, it remains sensitive to initialization and upstream relation-inference errors. In parallel, several approaches use generated images or videos as intermediate supervision and lift them back to 3D (Ling et al., 2025; Huang et al., 2025; Bian et al., 2025; Wang et al., 2024), improving visual realism but trading off precise instruction following and controllability. Crucially, most prior works resolve inconsistent or infeasible relations via post-hoc heuristics, such as grid-map discretization and pruning conflicting relations (Yang et al., 2024b; Sun et al., 2025), which can distort the original semantics. In contrast, we address relation-reasoning errors during the MLLM's relative spatial reasoning and propose targeted improvements that yield relations that are semantically consistent and physically feasible.

## 3. Method

### 3.1. Problem Formulation

We study instruction-driven 3D layout generation, which places a set of 3D assets in a space according to natural-

language instructions. Formally, given an instruction $I$, a space with known size $(L, W, H)$, and a set of $N$ 3D assets $\mathcal{A} = \{a_i\}_{i=1}^N$, the goal is to generate a 3D layout that satisfies the instruction while remaining physically feasible.

Following previous works (Yang et al., 2024b; Sun et al., 2025), we use assets from Objaverse (Deitke et al., 2023). We assume that each object is upright and is only allowed to rotate around $+z$ axis. We use GPT-4o (Hurst et al., 2024) to annotate each asset with a short textual description, and determine its front-facing orientation from four canonical rendered views. The object size $(l_i, w_i, h_i)$ is defined as the dimensions of its axis-aligned bounding box after rotating the asset to face $+y$ axis. The output layout $S$ specifies the object pose of each asset $p_i = (x_i, y_i, z_i, \theta_i)$, where $(x_i, y_i, z_i) \in \mathbb{R}^3$ denotes the object center position and $\theta_i \in \mathbb{R}$ denotes the rotation around $+z$ axis.

Notably, we restrict our study to floor objects to isolate the core challenge of relative spatial reasoning. Accordingly, we predict only the planar pose of each asset, $p_i = (x_i, y_i, \theta_i)$, and fix its height as $z_i = h_i/2$. Extending the pipeline to wall-mounted or small tabletop objects would mainly require additional attachment or support relations (Pun et al., 2026) and is beyond the scope of this paper.

Our framework, **R³L**, as illustrated in Figure 2, consists of two stages: *reasoning* and *solving*. Given an instruction, a 3D space and a set of 3D assets, we use an MLLM to infer a set of metricized relative spatial relations among assets. During the *reasoning* stage, we introduce two key modules, **invariant spatial decomposition** and **consistent spatial imagination**, to improve the semantic consistency and physical feasibility of the inferred relations in Sections 3.2 and 3.3. During the *solving* stage, we translate these relations into differentiable constraints and optimize them via **supportive spatial optimization** in Section 3.4.

### 3.2. Invariant Spatial Decomposition

As the number of assets and relations grows, the MLLM often struggles to maintain consistent intermediate states, leading to confusion and accumulated reasoning errors. To improve tractability, semantic grouping (Sun et al., 2025) has been introduced to group semantically related assets and generate relations group by group. However, despite reducing the problem scale, semantic grouping remains insufficient to effectively mitigate error accumulation, because it does not reduce the number of reference-frame transformations required for multi-hop relation inference.

To address this, we propose **invariant spatial decomposition**, which partitions a scene into a set of frame-invariant units such that the intra-unit relative configuration is preserved under reference-frame transformations. Each unit is defined by an anchor asset and its member assets. This de-

composition factorizes relations into *intra-unit* relations that specify relative spatial relations within each unit and *inter-unit* relations that capture relations among unit anchors and the scene. As a result, when inferring relations involving a unit member, the MLLM can reason locally in the anchor-defined frame via member-to-anchor transformations, while the anchor-to-global transformations are handled separately by the inter-unit relations. This reduces repeated global reference-frame transformations along multi-hop chains for unit members, thereby curbing semantic drift.

Formally, we partition $N$ assets $\mathcal{A} = \{a_i\}_{i=1}^N$ into $K$ frame-invariant units $\mathcal{U} = \{U_k\}_{k=1}^K$ via an assignment function:

$$\pi : \{1, \ldots, N\} \to \{0, 1, \ldots, K\} \quad (1)$$

where $\pi(i) = 0$ denotes independent asset and $\pi(i) = k$ indicates $a_i \in U_k$. We obtain $\pi$ by prompting the MLLM to identify strongly coupled spatial components and maintain their internal relative configuration as rigid units during subsequent reasoning. For each unit $U_k$, we choose an anchor $a_k^{\text{anchor}} \in U_k$ to define a unit-local frame for intra-unit reasoning. Once formed, each frame-invariant unit is treated as a rigid assembly and reason only over its global pose $P_k = (x_k, y_k, \theta_k)$. Each member asset $a_i \in U_k$ has a local pose $p_{i,k}$, and its global pose is obtained by:

$$p_i = P_{\pi(i)} \oplus p_{i,\pi(i)}, \quad \forall \pi(i) \neq 0, \quad (2)$$

where $\oplus$ denotes planar rigid transformation composition. We further denote independent assets as $\mathcal{A}^0 = \{a_i \in \mathcal{A} \mid \pi(i) = 0\}$. All $a_i \in \mathcal{A}^0$ are considered as atomic (*i.e.*, anchor-only) units, with poses $\{p_i\}_{a_i \in \mathcal{A}^0}$ defined in the global frame. Let $\bar{\mathcal{U}} = \mathcal{U} \cup \mathcal{A}^0$ denote this full unit set.

After all units are established, we generate relative spatial relations at two levels. First, the MLLM generates a set of intra-unit relations $\mathcal{R}_k^{\text{intra}}$ for each $U_k \in \mathcal{U}$, which specify relations among assets within its unit-local frame. Then, the MLLM generates a set of inter-unit relations $\mathcal{R}^{\text{inter}}$ which govern unit placement in the scene: $\mathcal{R}^{\text{inter}} \subseteq \bar{\mathcal{U}} \times (\bar{\mathcal{U}} \cup \{s\})$, where $s$ denotes the scene. This gives the full relation set:

$$\mathcal{R} = \left( \bigcup_{k=1}^K \mathcal{R}_k^{\text{intra}} \right) \cup \mathcal{R}^{\text{inter}}. \quad (3)$$

From a graph perspective, our two-level relation generation induces a directed relation graph $G = (V, E)$. The node set is $V = \mathcal{A} \cup \{s\}$. Each relation $r \in \mathcal{R}$ corresponds to a directed edge $(u \to v) \in E$, following a reference-to-target convention (*i.e.*, the pose of $v$ is constrained in the reference frame of $u$). Crucially, our decomposition introduces a *vertex cut* formed by unit anchors, which factorizes $G$ into local intra-unit subgraphs and an inter-unit graph defined

over $\bar{\mathcal{U}}$ and $s$. Concretely, the edge set $E$ is decomposed as:

$$E = \Big( \bigcup_{k=1}^{K} E_k^{\text{intra}} \Big) \cup E^{\text{inter}}, \qquad (4)$$

where $E_k^{\text{intra}}$ and $E^{\text{inter}}$ are induced by $\mathcal{R}_k^{\text{intra}}$ and $\mathcal{R}^{\text{inter}}$.

We define the number of reference-frame transformations along a reasoning chain as the number of intermediate reference-frame switches along its directed path:

$$\mathcal{T}_{\text{path}}(\gamma) = m - 1, \quad \gamma = (v_0 \to \cdots \to v_m). \qquad (5)$$

Under the reference-to-target convention, each intermediate node introduces one reference-frame switch, so $\mathcal{T}_{\text{path}}(\gamma)$ counts the total number of reference-frame transformations along $\gamma$. With our vertex-cut factorization, inferring relations involving a unit member starts from its unit anchor rather than the scene node $s$, avoiding the shared prefix $s \to a_k^{\text{anchor}}$ that would otherwise be traversed repeatedly for every member. Thus, the cumulative number of reference-frame transformations in member-related inference is reduced. A formal statement is deferred to Appendix A.

### 3.3. Consistent Spatial Imagination

Inferring reliable metric displacements in metricized relations is intrinsically difficult for MLLMs, since it requires an implicit global space allocation of all assets, rather than independent pairwise guesses. Without an explicit spatial representation to verify these numeric offsets, MLLMs can produce displacements that are locally plausible yet globally incompatible. This issue is exacerbated under multi-hop reasoning, where implied displacements are composed across changing reference frames. As a result, small per-hop errors can accumulate into metric drift, leading to inconsistent spacing, collisions, and physically infeasible layouts.

To address this, we propose **consistent spatial imagination** to promote self-consistency of the predicted metricized relations. Motivated by (Tolman, 1948; Yang et al., 2025a), we construct global-local cognitive maps based on our frame-invariant units: local maps stabilize intra-unit configurations and the global map allocates space among units and independent assets. Leveraging these maps, we guide the MLLM to perform an *imagine-and-revise* loop: it imagines object placements in both the unit-level local frame and the global frame, detects geometric violations (*e.g.*, collisions), and iteratively revises the relations responsible for these conflicts. This process grounds metricized relations in explicit scene geometry, suppressing metric drift and yielding more feasible and self-consistent relations.

We implement global-local cognitive maps as explicit spatial representations that the MLLM is required to externalize during spatial reasoning. Formally, for each frame-invariant

unit $U_k$, the MLLM estimates local poses $q_{i,k}$ under the unit-local map from intra-unit relations $\mathcal{R}_k^{\text{intra}}$ and the poses of each unit $Q_k$ together with the poses of independent assets $\{q_i\}_{a_i \in \mathcal{A}^0}$ under the global map from inter-unit relations $\mathcal{R}^{\text{inter}}$. The cognitive maps are defined as:

$$\begin{aligned} \mathcal{M}_k^{\text{local}} &= \{q_{i,k}\}_{a_i \in \mathcal{A}(U_k)}, \\ \mathcal{M}^{\text{global}} &= \{Q_k\}_{U_k \in \mathcal{U}} \cup \{q_i\}_{a_i \in \mathcal{A}^0}. \end{aligned} \qquad (6)$$

where the global pose $q_i$ for each $a_i \in U_{\pi(i)}$ is induced by $Q_{\pi(i)}$ and $q_{i,\pi(i)}$ via the same rigid transformation composition as in Equation (2). For an asset, $(l_i, w_i)$ is its canonical planar size, while for a unit, $(l_i, w_i)$ is defined as the planar size of the unit's axis-aligned bounding box. To facilitate rotation-aware reasoning and collision detection, we additionally require the MLLM to derive two auxiliary geometric fields. First, it computes the axis-aligned extents of the yaw-rotated planar footprint, $(e_i^x(\theta_i), e_i^y(\theta_i))$, as

$$\begin{aligned} e_i^x(\theta_i) &= |l_i \cos\theta_i| + |w_i \sin\theta_i|, \\ e_i^y(\theta_i) &= |l_i \sin\theta_i| + |w_i \cos\theta_i|. \end{aligned} \qquad (7)$$

Second, it derives the axis-aligned bounds induced by footprint extents $(e_i^x(\theta_i), e_i^y(\theta_i))$ and center position $(x_i, y_i)$,

$$\begin{aligned} B_i^x &= \Big[ x_i - \tfrac{1}{2} e_i^x(\theta_i),\ x_i + \tfrac{1}{2} e_i^x(\theta_i) \Big], \\ B_i^y &= \Big[ y_i - \tfrac{1}{2} e_i^y(\theta_i),\ y_i + \tfrac{1}{2} e_i^y(\theta_i) \Big], \end{aligned} \qquad (8)$$

where $B_i = (B_i^x, B_i^y)$. Together, these poses, footprint extents $(e_i^x, e_i^y)$, and bounds $B_i$ form an explicit spatial representation that supports lightweight consistency checks at both local and global levels. In particular, two assets or units $i \neq j$ are considered colliding if their axis-aligned bounds have positive overlap on both axes:

$$\text{Collide}(i, j) \iff |B_i^x \cap B_j^x| > 0 \wedge |B_i^y \cap B_j^y| > 0, \qquad (9)$$

where $|\cdot|$ denotes the interval length. Note that this bound-based consistency check serves only as a lightweight reasoning proxy for global simulation. When unit overlaps are ambiguous under this proxy, the MLLM can further refer to member-level bounds for refinement.

The local-to-global imagine-and-revise loop is carried out implicitly within the MLLM's reasoning to promote self-consistency. At iteration $t$, the MLLM internally instantiates cognitive maps $\mathcal{M}_k^{\text{local},(t)} \cup \mathcal{M}^{\text{global},(t)}$ from the current relation set $\mathcal{R}^{(t)}$, evaluates collisions via $\text{Collide}(\cdot, \cdot)$, and performs localized revision, which edits intra-unit relations for intra-unit collisions and inter-unit relations for inter-unit collisions. The MLLM then obtains the updated relation set $\mathcal{R}^{(t+1)}$ and repeats the simulation until no collisions are detected or the model exhausts its internal reasoning budget.

## 3.4. Supportive Spatial Optimization

Given the inferred metricized relative relations, we follow (Sun et al., 2025) to translate them into differentiable constraints and optimize object poses. However, jointly optimizing all objects is often unstable and converges slowly. The key reason is that constraints are unevenly distributed across objects: some objects are constrained by many others, while some are constrained by only a few. Moving a highly constrained object changes many constraint terms simultaneously, so fixing one violation can trigger new ones elsewhere, causing oscillatory updates and slow convergence.

Prior work alleviates this issue by optimizing group by group (Sun et al., 2025), which simplifies optimization but makes early placements difficult to revise. In contrast, we design **supportive spatial optimization** that stabilizes optimization via global-to-local pose re-parameterization, while allowing the full scene to be jointly optimized.

Specifically, we optimize object poses in a mixed representation $\tilde{p}$ via global-to-local re-parameterization. Independent assets are optimized in the global frame with poses $p_i = (x_i, y_i, \theta_i)$. For each unit $U_k$, we optimize a unit-level pose $P_k = (x_k, y_k, \theta_k)$ in the global frame, while optimizing each member in the unit-local frame using a local pose $p_i^\ell = (x_i^\ell, y_i^\ell, \theta_i^\ell)$ instead of a global pose. Its global pose is recovered from $(P_k, p_i^\ell)$ via Equation (2). This reparameterization decouples intra-unit gradients from unit poses (Proposition B.1), thereby simplifying optimization. We provide a structural analysis in Appendix B.

Let $\tilde{p}$ denote independent asset poses, unit poses $\{P_k\}_{k=1}^K$, and member local poses $\{p_i^\ell \mid \pi(i) \neq 0\}$. We instantiate each relation $r \in \mathcal{R}$ as a differentiable penalty $\ell(r; \tilde{p})$ that is zero when the relation is satisfied and increases with the violation magnitude. $\mathcal{L}_{\text{rel}}^k$ and $\mathcal{L}_{\text{rel}}^{\text{global}}$ are obtained by summing $\ell(r; \tilde{p})$ over relations in $\mathcal{R}_k^{\text{intra}}$ and $\mathcal{R}^{\text{inter}}$.

**Local term.** For each unit $U_k$, $\mathcal{L}_{\text{local}}^k$ is defined in its unit-local frame, consisting of (i) member-level collision losses, and (ii) intra-unit relational losses induced by $\mathcal{R}_k^{\text{intra}}$:

$$\mathcal{L}_{\text{local}}^k(\tilde{p}) = \lambda_{\text{col}}\mathcal{L}_{\text{col}}^k(\tilde{p}) + \lambda_{\text{rel}}\mathcal{L}_{\text{rel}}^k(\tilde{p}) \quad (10)$$

where $\lambda_{\text{col}}$ and $\lambda_{\text{rel}}$ are scalar weights.

**Global term.** $\mathcal{L}_{\text{global}}$ evaluates inter-unit constraints among independent assets and units. We represent each unit $U_k$ as an axis-aligned bounding box (AABB) that tightly encloses all member boxes in the unit frame, and transform it by the unit pose $P_k$ into an oriented bounding box (OBB). We then apply (i) boundary losses to keep independent assets and unit OBBs inside the room, (ii) collision losses among independent objects and unit OBBs, and (iii) inter-entity relational losses induced by $\mathcal{R}^{\text{inter}}$:

$$\mathcal{L}_{\text{global}}(\tilde{p}) = \lambda_{\text{bd}}\mathcal{L}_{\text{bd}}(\tilde{p}) + \lambda_{\text{col}}\mathcal{L}_{\text{col}}^{\text{global}}(\tilde{p}) + \lambda_{\text{rel}}\mathcal{L}_{\text{rel}}^{\text{global}}(\tilde{p}) \quad (11)$$

where $\lambda_{\text{bd}}$, $\lambda_{\text{col}}$, and $\lambda_{\text{rel}}$ are scalar weights.

Overall, we minimize a two-level objective over $\tilde{p}$:

$$\mathcal{L}(\tilde{p}) = \mathcal{L}_{\text{global}}(\tilde{p}) + \sum_{k=1}^K \mathcal{L}_{\text{local}}^k(\tilde{p}). \quad (12)$$

## 4. Experiments

In our experiments, we seek to answer the following questions: **Q1:** How does R³L compare with existing instruction-driven 3D layout generation baselines? **Q2:** How essential are invariant spatial decomposition and consistent spatial imagination for inferring reliable and consistent metricized relative spatial relations? **Q3:** How does supportive spatial optimization affect optimization stability and convergence speed over successive optimization steps?

**Settings.** We evaluate R³L against existing methods under open-vocabulary instructions across diverse scene types. Following (Sun et al., 2025; Yang et al., 2025b), we create test cases across 9 scene types with 3 scenes per type, and design instructions at three difficulty levels, from easy to hard. Each test case contains up to 40 floor-standing assets preprocessed from Objaverse (Deitke et al., 2023). All methods are evaluated on the same instructions and preprocessed assets. We use GPT-5 in all experiments. Details of test case generation are provided in Appendix G.1.

**Baselines.** We compare R³L with three recent approaches for instruction-driven 3D layout generation: LayoutGPT (Feng et al., 2023), Holodeck (Yang et al., 2024b), and LayoutVLM (Sun et al., 2025). LayoutGPT directly predicts absolute object poses via in-context examples. Holodeck first generates relational constraints and then solves them with a DFS-based solver. LayoutVLM predicts numerical object poses with spatial relations, and refines the layout via differentiable optimization. Since LayoutGPT was originally limited to bedrooms and living rooms, we adapt it to open-vocabulary instructions for a fair comparison.

**Metrics.** Following (Sun et al., 2025; Yang et al., 2025b; Ling et al., 2025), we evaluate the generated 3D layouts using both physical and semantic metrics. For physical evaluation, we report the collision ratio (%CR) and the out-of-bound ratio (%OR), defined as the percentage of objects involved in any collision and the percentage of out-of-bound objects, where lower values indicate fewer physical violations. For semantic evaluation, we measure realism (Real.) and functionality (Func.), instruction following (Instr.), where Instr. measures how well the layout matches the instruction. Following (Çelen et al., 2024; Ling et al., 2025), we adopt Gemini 3 Flash as an LLM-based evaluator: given the instruction and two renderings of the generated scene (top-down and side views), it scores each semantic metric from 1 to 10, where higher is better. For pair-wise com-

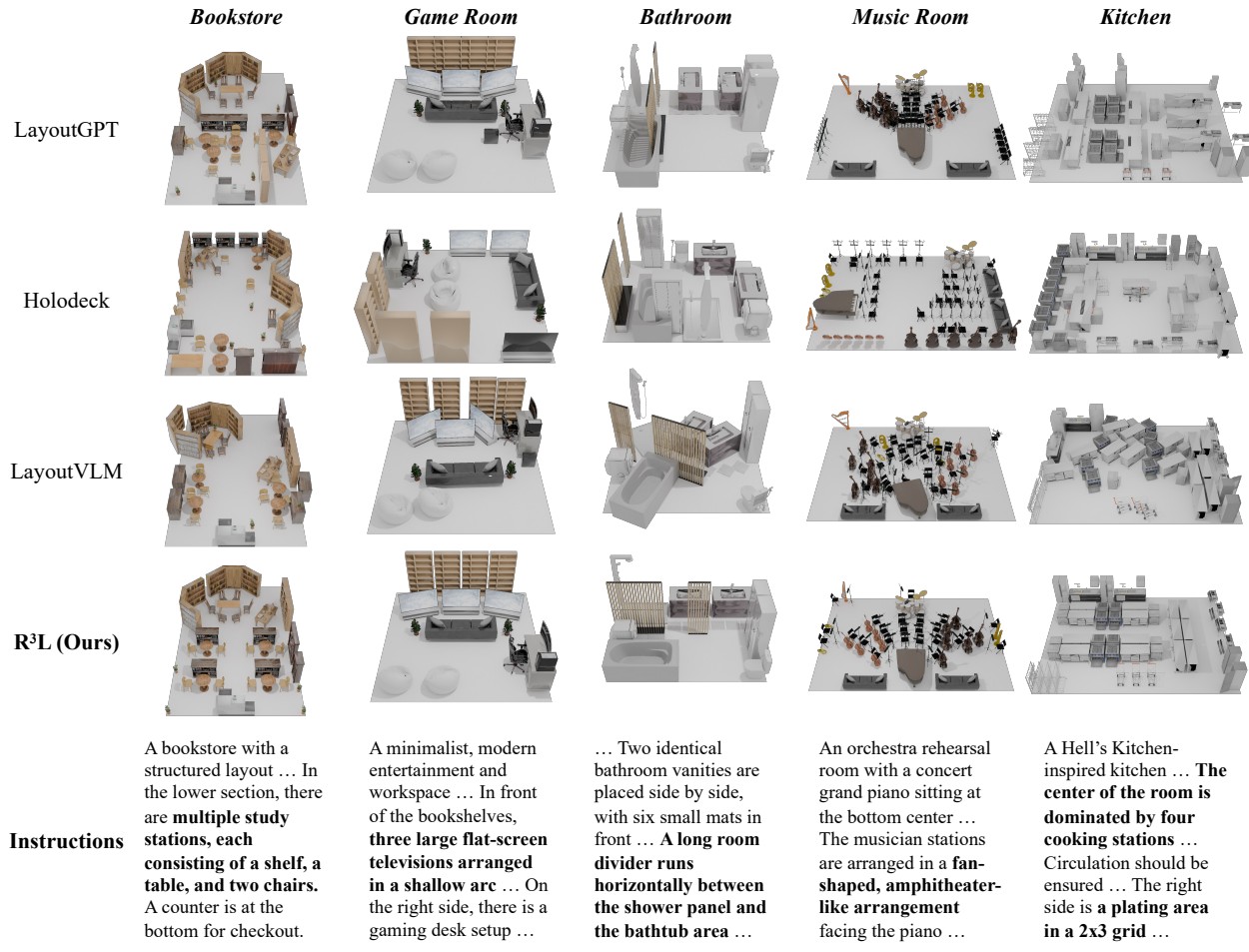

|  | Bookstore | Game Room | Bathroom | Music Room | Kitchen |
|--|--|--|--|--|--|

**Instructions**

A bookstore with a structured layout … In the lower section, there are **multiple study stations, each consisting of a shelf, a table, and two chairs.** A counter is at the bottom for checkout.

A minimalist, modern entertainment and workspace … In front of the bookshelves, **three large flat-screen televisions arranged in a shallow arc** … On the right side, there is a gaming desk setup …

… Two identical bathroom vanities are placed side by side, with six small mats in front … **A long room divider runs horizontally between the shower panel and the bathtub area** …

An orchestra rehearsal room with a concert grand piano sitting at the bottom center … The musician stations are arranged in a **fan-shaped, amphitheater-like arrangement** facing the piano …

A Hell's Kitchen-inspired kitchen … **The center of the room is dominated by four cooking stations** … Circulation should be ensured … The right side is **a plating area in a 2x3 grid** …

*Figure 3.* Qualitative comparison between R³L and existing methods for generating layouts based on language instructions. R³L demonstrates strong instruction-following ability while maintaining physically plausible and visually coherent layouts.

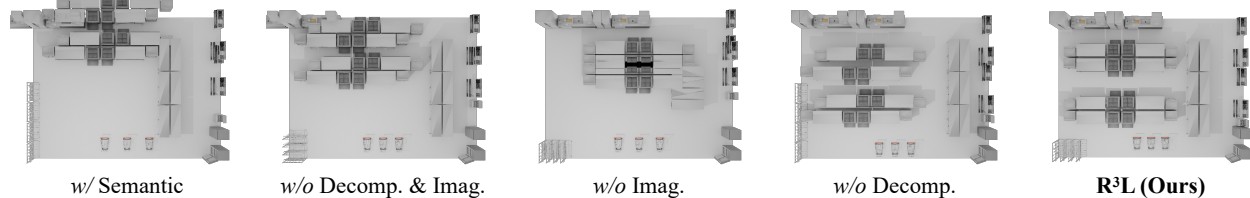

*w/ Semantic*   *w/o Decomp. & Imag.*   *w/o Imag.*   *w/o Decomp.*   **R³L (Ours)**

*Figure 4.* Visual ablation of Semantic, Decomp., and Imag. Physical losses are disabled to isolate the effect of relation inference.

parisons, we report the win rate and the Elo rating, where higher indicates stronger overall preference.

## 4.1. Evaluation on 3D Layout Generation

We present the quantitative results in Table 1. R³L achieves the strongest semantic performance while maintaining physical feasibility, consistently outperforming all baselines across the nine scene types. Compared with LayoutGPT, the strongest semantic baseline, R³L improves layout coherence by 2.0 and instruction following by 1.0, while reducing the average collision and out-of-bound ratios from 4.3%

and 8.1% to 0.0% and 0.0%, respectively. This reflects LayoutGPT's limitation: directly predicting absolute object poses often yields semantically plausible but physically invalid layouts. Compared with Holodeck, R³L matches its strong physical feasibility while substantially improving semantic performance, including a +6.1 gain in instruction following. This stems from Holodeck's reliance on coarse relation modeling and grid-based solving, which favor feasibility but often weaken semantic fidelity. LayoutVLM better balances semantics and feasibility through differentiable optimization, but remains sensitive to inaccurate or conflicting

*Table 1.* Quantitative results for instruction-driven 3D layout generation across nine scene types. Lower is better for %CR/%OR, and higher is better for Real./Func./Instr. R$^3$L achieves the best overall physical feasibility and semantic performance.

| Method | Bathroom | | | | | Bedroom | | | | | Bookstore | | | | | Game Room | | | | | Gym | | | | |
|---|---|---|---|---|---|---|---|---|---|---|---|---|---|---|---|---|---|---|---|---|---|---|---|---|---|
| | %CR | %OR | Real. | Func. | Instr. | %CR | %OR | Real. | Func. | Instr. | %CR | %OR | Real. | Func. | Instr. | %CR | %OR | Real. | Func. | Instr. | %CR | %OR | Real. | Func. | Instr. |
| LayoutGPT | 7.6 | 12.1 | 5.9 | 5.3 | 7.9 | 8.7 | 0.0 | 4.4 | 3.9 | 7.7 | 2.9 | 2.1 | 7.5 | 7.0 | 8.1 | 5.1 | 17.5 | 5.1 | 5.1 | 8.4 | 7.4 | 25.0 | 6.5 | 6.3 | 7.3 |
| Holodeck | 4.0 | 0.0 | 2.9 | 2.3 | 1.9 | 0.0 | 0.0 | 4.5 | 4.0 | 3.9 | 0.6 | 0.0 | 4.5 | 3.7 | 3.1 | 0.6 | 0.0 | 3.9 | 3.0 | 2.1 | 1.7 | 1.7 | 5.9 | 5.6 | 5.9 |
| LayoutVLM | 3.0 | 13.2 | 3.5 | 3.5 | 4.7 | 0.3 | 6.8 | 6.4 | 5.9 | 7.3 | 1.1 | 7.3 | 3.4 | 4.3 | 5.5 | 0.1 | 7.7 | 6.3 | 5.4 | 8.7 | 0.1 | 29.4 | 4.8 | 5.6 | 6.7 |
| R$^3$L (Ours) | 0.0 | 0.0 | 7.5 | 7.5 | 9.4 | 0.0 | 0.0 | 6.9 | 6.5 | 7.9 | 0.0 | 0.0 | 8.9 | 8.9 | 8.9 | 0.0 | 0.0 | 7.3 | 6.5 | 8.7 | 0.0 | 0.0 | 8.6 | 8.0 | 9.7 |
| Method | Kitchen | | | | | Living Room | | | | | Music Room | | | | | Restaurant | | | | | Average | | | | |
| | %CR | %OR | Real. | Func. | Instr. | %CR | %OR | Real. | Func. | Instr. | %CR | %OR | Real. | Func. | Instr. | %CR | %OR | Real. | Func. | Instr. | %CR | %OR | Real. | Func. | Instr. |
| LayoutGPT | 1.8 | 7.0 | 6.5 | 6.5 | 8.2 | 1.6 | 0.0 | 5.2 | 4.7 | 7.9 | 1.7 | 0.5 | 6.8 | 6.5 | 8.9 | 1.8 | 8.4 | 5.0 | 4.5 | 8.1 | 4.3 | 8.1 | 5.9 | 5.5 | 8.1 |
| Holodeck | 1.5 | 0.8 | 4.0 | 3.0 | 2.5 | 2.8 | 2.1 | 3.1 | 2.7 | 2.7 | 0.7 | 2.4 | 3.9 | 2.6 | 2.1 | 0.1 | 0.0 | 2.9 | 2.5 | 2.7 | 1.3 | 0.8 | 4.0 | 3.3 | 3.0 |
| LayoutVLM | 0.5 | 0.0 | 5.5 | 5.1 | 6.9 | 0.1 | 16.9 | 5.4 | 5.0 | 8.0 | 0.1 | 14.7 | 4.4 | 4.8 | 7.3 | 0.1 | 2.4 | 3.9 | 3.5 | 6.4 | 0.6 | 10.9 | 4.9 | 4.8 | 6.9 |
| R$^3$L (Ours) | 0.0 | 0.0 | 8.7 | 8.3 | 9.5 | 0.0 | 0.0 | 7.8 | 7.1 | 9.6 | 0.4 | 0.0 | 7.9 | 7.7 | 8.6 | 0.0 | 0.0 | 7.5 | 7.0 | 9.1 | 0.0 | 0.0 | 7.9 | 7.5 | 9.1 |

*Table 2.* Pairwise comparison results among R$^3$L and the other baselines. Win rates show the number of tasks in which the row method is preferred over the column method across 27 tasks. Higher Elo scores indicate stronger overall performance.

| Method | Win Rates | | | | Elo ↑ |
|---|---|---|---|---|---|
| | LayoutGPT | Holodeck | LayoutVLM | R$^3$L | |
| LayoutGPT | – | 25/27 | 21/27 | 4/27 | 1597 |
| Holodeck | 2/27 | – | 4/27 | 0/27 | 1117 |
| LayoutVLM | 6/27 | 23/27 | – | 3/27 | 1419 |
| R$^3$L (Ours) | 23/27 | 27/27 | 24/27 | – | 1866 |

*Table 3.* **Ablation Studies.** We ablate the two key modules of R$^3$L. We also compare against semantic grouping (Sun et al., 2025). For fair comparison, we disable physical losses so that layouts are determined solely by the inferred metricized relations.

| Method | %CR | %OR | Real. | Func. | Instr. |
|---|---|---|---|---|---|
| *w/* Semantic | 3.2 | 9.7 | 5.3 | 5.1 | 7.5 |
| *w/o* Decomp. & Imag. | 3.1 | 7.9 | 5.7 | 5.4 | 7.8 |
| *w/o* Imag. | 3.1 | 6.3 | 6.2 | 6.0 | 7.9 |
| *w/o* Decomp. | 1.7 | 3.8 | 6.9 | 6.7 | 8.5 |
| **R$^3$L (Ours)** | **1.0** | **1.6** | **8.0** | **7.3** | **9.1** |

inferred relations, leading to lower semantic scores. Pairwise comparisons in Table 2 further demonstrate that R$^3$L is preferred in over 85% of scenes when compared with the other baselines and achieves the highest Elo rating. Together, these results highlight the effectiveness of R$^3$L in generating layouts that are both physically feasible and semantically consistent, and faithful to the instructions.

We also provide qualitative comparisons in Figure 3. R$^3$L demonstrates strong instruction-following ability while maintaining physically plausible and visually coherent layouts. For example, in the *Bookstore* case, only R$^3$L correctly captures the requirement of "multiple study stations", whereas other methods either fail to satisfy this requirement or produce physically implausible layouts.

### 4.2. Additional Analyses

**Ablation Studies.** We ablate the two key modules of R$^3$L: invariant spatial decomposition (Decomp.) and consistent spatial imagination (Imag.). We also compare against semantic grouping (Semantic) (Sun et al., 2025). Specifically, we consider four variants: 1) *w/* Semantic, 2) *w/o* Decomp. & Imag., 3) *w/o* Imag., and 4) *w/o* Decomp. To isolate the effect of relation inference, we optimize layouts using only relational losses from the inferred metricized relations, with physical losses disabled to avoid confounding effects.

As shown in Table 3 and Figure 4, removing both Decomp. and Imag. leads to a substantial drop in both semantic qual-

ity and physical plausibility. Semantic grouping performs worst overall, suggesting that a purely semantics-driven decomposition is insufficient to stabilize relative spatial reasoning. Using Decomp. without Imag. improves semantic consistency, but often yields physically implausible layouts because metric composition errors remain uncorrected. Conversely, using Imag. without Decomp. improves physical plausibility, but degrades semantic consistency due to frequent reference-frame transformations in long relation chains. Overall, Decomp. and Imag. are complementary: Decomp. reduces reference-frame transformations and stabilizes multi-hop semantics, while Imag. promotes self-consistency and suppresses metric drift. Combining both yields the strongest overall performance.

**Placement Error Rate by Hop Count.** To directly evaluate the effect of reference-frame transformations on relative spatial reasoning, we analyze placement errors with respect to hop count. As shown in Figure 5, the error rate increases substantially with hop count when both modules are removed. Adding decomposition flattens this growth trend by reducing reference-frame transformations, while imagination lowers the overall error level by improving global consistency. Combining both, the full model achieves consistently low error rates across all hop counts, indicating reduced error accumulation under long relation chains. The detailed results are provided in Table 15.

**User Study.** To further evaluate the quality of scenes generated by R$^3$L and assess the agreement between Gemini

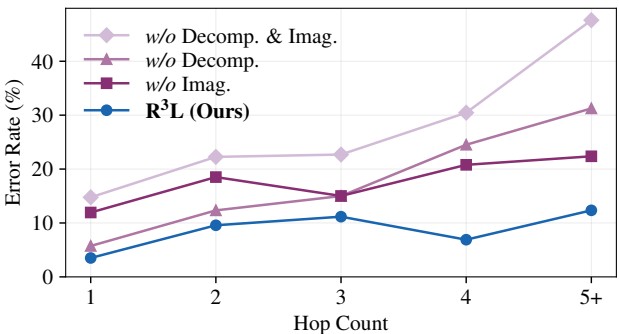

*Figure 5.* **Placement Error Rate by Hop Count.** A larger hop count implies more reference-frame transformations along the inferred relation chain. Decomposition flattens the growth of error with hop count, while imagination lowers the overall error level. Combining both yields the lowest error rates across all hops.

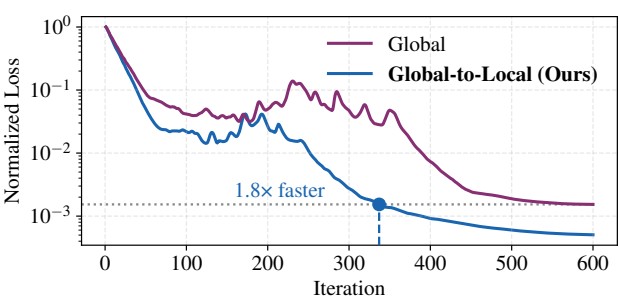

*Figure 6.* Comparison of convergence curves for global and global-to-local optimization under the same optimizer settings. Our global-to-local optimization converges faster and maintains lower loss throughout most of the optimization.

*Table 4.* User study results on semantic scores. Entries are reported as mean $\pm 2$ standard errors of the mean (SEM).

| Method | Real. | Func. | Instr. |
|---|---|---|---|
| LayoutGPT | $4.1 \pm 0.4$ | $4.1 \pm 0.4$ | $5.7 \pm 0.4$ |
| Holodeck | $5.3 \pm 0.3$ | $5.2 \pm 0.3$ | $4.3 \pm 0.3$ |
| LayoutVLM | $2.8 \pm 0.3$ | $3.3 \pm 0.3$ | $5.0 \pm 0.4$ |
| **R³L (Ours)** | $\mathbf{6.7 \pm 0.3}$ | $\mathbf{6.5 \pm 0.3}$ | $\mathbf{7.9 \pm 0.3}$ |

*Table 5.* User study results of pairwise comparisons among all methods. Some win counts do not sum to 27 because ties are allowed. Elo ratings are computed over all non-tied pairs.

| Method | Win Rates | | | | Elo ↑ |
|---|---|---|---|---|---|
| | LayoutGPT | Holodeck | LayoutVLM | R³L | |
| LayoutGPT | – | 21/27 | 21/27 | 1/27 | 1511 |
| Holodeck | 6/27 | – | 11/27 | 1/27 | 1276 |
| LayoutVLM | 4/27 | 15/27 | – | 2/27 | 1308 |
| **R³L (Ours)** | **25/27** | **26/27** | **25/27** | – | **1905** |

*Table 6.* **Human-LLM Alignment.** Kendall's $\tau$-b between rankings induced by human judgments and automatic scores. User-User denotes inter-human agreement, while User-LLM denotes human-LLM agreement. Higher values indicate stronger agreement.

| Method | Real. ↑ | Func. ↑ | Instr. ↑ |
|---|---|---|---|
| User-User | **0.56** | **0.55** | 0.53 |
| User-LLM | 0.52 | 0.50 | **0.56** |

re-parameterization improves optimization efficiency by reducing gradient coupling during updates. The complete set of convergence curves is provided in Figure 16.

## 5. Conclusion

In this paper, we propose R³L, a general framework that improves the reliability and consistency of multi-hop relative spatial reasoning for 3D layout generation. To mitigate semantic and metric drift, we present invariant spatial decomposition to reduce reference-frame transformations, consistent spatial imagination to promote self-consistent relation inference, and supportive spatial optimization to improve optimization efficiency while preserving global coherence. Experiments show that R³L consistently produces physically feasible, semantically consistent, and instruction-faithful layouts. Future work will extend R³L to broader spatial reasoning tasks beyond 3D layout generation.

## Acknowledgments

This work was jointly supported by the Young Scientists Fund of the Research Grants Council of Hong Kong (25206524, 15212925) and the National Natural Science Foundation of China (42301520).

3 Flash ratings and human judgments, we conduct a user study with 150 participants. As reported in Table 4, R³L achieves the best performance across all dimensions. We also conduct a pairwise preference study, in which participants choose between scenes generated by R³L and each baseline. As summarized in Table 5, R³L is preferred in more than 90% of comparisons. Overall, the human study corroborates our automatic evaluation and confirms that R³L produces visually plausible and semantically consistent scenes. Full study details are provided in Appendix H.

**Human-LLM Alignment.** As shown in Table 6, we compute Kendall's $\tau$-b between rankings induced by human judgments and automatic scores. The correlations are positive across all three semantic metrics and close to the User-User agreement levels, indicating that LLM-based evaluation is broadly consistent with human judgment.

**Convergence Speed.** As shown in Figure 6, our supportive spatial optimization (*i.e.*, global-to-local optimization) converges faster than standard global optimization under the same settings (*e.g.*, learning rates), achieving an average $1.8\times$ speedup and maintaining lower loss throughout most of the optimization. This indicates that our global-to-local

## Impact Statement

This paper presents work whose goal is to advance the field of Machine Learning. There are many potential societal consequences of our work, none of which we feel must be specifically highlighted here.

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

The appendix includes:

*Table 7.* Definitions of notation used in Appendices A and B

| Notation | Meaning |
|---|---|
| $G = (V, E)$ | Relation graph w/ nodes $V$ and edges $E$. |
| $s$ | Scene-level reference node. |
| $d(u, v)$ | Directed shortest-path length $u \to v$. |
| $U_k$ | The $k$-th spatial unit. |
| $a_k$ | Anchor asset of unit $U_k$. |
| $M_k$ | Non-anchor members of $U_k$. |
| $v \in M_k$ | A non-anchor member. |
| $|M_k|$ | Number of non-anchor members. |
| $P_k$ | Global pose of unit $U_k$. |
| $a_i, a_j$ | Assets in the same unit. |
| $p_i, p_j$ | Global poses of assets $a_i, a_j$. |
| $p_i^\ell, p_j^\ell$ | Local poses in the unit frame. |
| $\oplus$ | Pose composition. |
| $p_i^{-1} \oplus p_j$ | Relative pose from $a_i$ to $a_j$. |
| $\tilde{p}$ | Mixed global–local poses. |
| $r$ | Spatial relation constraint. |
| $\ell(r; \tilde{p})$ | Penalty for relation $r$. |
| $\varphi(\cdot)$ | Relation-specific penalty. |
| $\mathcal{R}_k^{\text{intra}}$ | Set of intra-unit relations involving $U_k$. |
| $\mathcal{R}_k^{\text{inter}}$ | Set of inter-unit relations involving $U_k$. |
| $n_k^{\text{intra}}$ | Number of intra-unit losses involving $U_k$ |
| $n_k^{\text{inter}}$ | Number of inter-unit losses involving $U_k$ |
| $\nabla_{P_k}$ | Gradient w.r.t. unit pose $P_k$. |
| $\beta$ | Fixed per-edge smoothness constant. |

## A. Proof of Reference Frame Shift Reduction

We formalize the claim used in Section 3.2: when a scene graph is factorized through a cut vertex, it strictly reduces the total number of reference frame shifts during multi-hop reasoning. The result is definitional, but we include it nonetheless to make the reduction factor explicit.

**Setup.** Following the notations in Section 3.2, the minimum number of frame shifts required from $u$ to $v$ is

$$T^*(u, v) = \min_{\gamma: u \to v} T_{\text{path}}(\gamma) = d(u, v) - 1, \quad u \neq v \quad (13)$$

where $d(u, v)$ denotes the shortest directed path length from $u \to v$. Therefore, the baseline cumulative cost for reasoning about all nodes from the scene node $s$ is:

$$\text{Cost}(G) = \sum_{v \in V} T^*(s, v) \quad (14)$$

**Proposition A.1** (Anchor-local reasoning reduces total transformations). *For a frame-invariant unit $U_k$ with anchor asset $a_k^{anchor}$, let $M_k = U_k \setminus \{a_k^{anchor}\}$ be a nonempty set of its members. Suppose that every directed path from $s$ to $v \in M_k$ passes through $a_k^{anchor}$ (i.e., $a_k^{anchor}$ is a cut vertex of $G$). Then, by reasoning about all such members from $a_k^{anchor}$ instead of $s$ using Invariant Spatial Decomposition, the cumulative cost becomes*

$$\text{Cost}'(G) = \sum_{v \in V \setminus M_k} T^*(s, v) + \sum_{v \in M_k} T^*(a_k^{\text{anchor}}, v) \quad (15)$$

*and the frame shift reduction $\Delta$ is precisely*

$$\Delta = \text{Cost}(G) - \text{Cost}'(G) = |M_k| \cdot d(s, a_k^{\text{anchor}}) \quad (16)$$

*Proof.* Fix any $v \in M_k$. Since every directed path from s to v passes through $a_k^{\text{anchor}}$, any shortest path from $s \to v$ decomposes at $a_k^{\text{anchor}}$, giving

$$d(s, v) = d(s, a_k^{\text{anchor}}) + d(a_k^{\text{anchor}}, v) \quad (17)$$

Therefore, the reduction $\Delta$ is

$$\begin{aligned}
\Delta &= \text{Cost}(G) - \text{Cost}'(G) \\
&= \sum_{v \in M_k} \left( T^*(s, v) - T^*(a_k^{\text{anchor}}, v) \right) \\
&= \sum_{v \in M_k} \left( (d(s, v) - 1) - (d(a_k^{\text{anchor}}, v) - 1) \right) \\
&= \sum_{v \in M_k} d(s, a_k^{\text{anchor}}) \\
&= |M_k| \cdot d(s, a_k^{\text{anchor}})
\end{aligned} \quad (18)$$

Since $a_k^{\text{anchor}} \neq s$ and reachable from $s$, $d(s, a_k^{\text{anchor}}) \geq 1$. Along with $|M_k| \geq 1$, the reduction is strict. $\square$

### A.1. Discussion

Every node $v \in M_k$ lies behind the same anchor $a_k^{\text{anchor}}$ in the relation graph. Without our decomposition, reasoning about every such member $v$ requires repeatedly traversing the same shared prefix from $s \to a_k^{\text{anchor}}$. Frame-invariant reasoning removes this redundant prefix, saving exactly $d(s, a_k^{\text{anchor}})$ frame shifts per member.

# B. Structural Analysis of Gradient Decoupling

This section provides a structural analysis of why the global-to-local re-parameterization introduced in Section 3.4 improves optimization stability, as empirically observed in Figure 6. The key observation is that even if each individual constraint term is well-behaved, graph topology can amplify loss stiffness when a variable participates in many terms. Our re-parameterization removes a specific high-degree curvature accumulation source.

## B.1. Motivation: Topological Gradient Coupling

Our optimization objective in Equation (12) sums differentiable relational penalties over the relation set $\mathcal{R}$

$$\mathcal{L}(\tilde{p}) = \sum_{r \in \mathcal{R}} \ell(r; \tilde{p}) \qquad (19)$$

where $\ell(r; \tilde{p})$ is the penalty for relation $r$, $\tilde{p}$ denotes the unified pose representation defined in Section 3.4, comprising independent asset poses $\{p_i\}_{a_i \in A_0}$, unit poses $\{P_k\}_{k=1}^K$, and member local poses $\{p_i^\ell \mid \pi(i) \neq 0\}$. The conditioning of said objective (i.e., how rapidly the gradient of $\mathcal{L}$ can change when a single pose is perturbed) governs the stability and convergence speed of our optimization. Two factors determine this conditioning:

**(i) Loss Function Design.** Given any fixed pair of nodes $(u, v)$ on the relation graph, the penalty $\ell(r; \tilde{p})$ can be linear, quadratic, hinge-type, or any other differentiable form. These choices control the *per-edge stiffness* of our objective: how rapidly the gradient of a *single* edge penalty term can change (i.e., the smoothness constant).

**(ii) Topological Coupling.** The relation graph's topology specifies which poses participate in which loss terms. When a single pose participates in many edges (e.g., an anchor connected to many members), the stiffness contributed by those edges can *accumulate* along certain directions, creating ill-conditioning even if the individual penalty terms are well-behaved.

**Which to isolate.** The global-to-local re-parameterization technique proposed in 3.4 is purely topological: it does not alter the functional form of any individual penalty $\ell(r; \tilde{p})$; instead, it changes which pose variables participate in which penalty terms. Our global-to-local re-parameterization benefits optimization through improving factor (ii), not (i).

**How to isolate.** To isolate the topological effect in this analysis, we must fix a uniform per-edge stiffness budget. Suppose that each penalty $\ell(r; \tilde{p})$ has block-gradient Lipschitz constant $\beta$ with respect to every pose block on which it depends (i.e., block-$\beta$-smooth). Then, given any pose block $\tilde{p}_u$ (i.e., $p_i$, $P_k$, or $p_i^\ell$) touched by $\ell(r; \tilde{p})$ and any perturbation $\delta$ supported on that block

$$\|\nabla_{\tilde{p}_u} \ell(r; \tilde{p} + \delta) - \nabla_{\tilde{p}_u} \ell(r; \tilde{p})\|_2 \leq \beta \|\delta\|_2 \qquad (20)$$

where $\beta$ is independent of graph degree. If pose block $\tilde{p}_u$ participates in $n$ penalty terms, then by triangle inequality, the block smoothness constant of $\mathcal{L}$ with respect to $\tilde{p}_u$ is at most $n\beta$. In other words, the block-wise smoothness upper bound grows linearly with the number of incident relations.

Under naive global parameterization, a unit anchor $a_k^{\text{anchor}}$ may participate in both $n_k^{\text{intra}}$ intra-unit relations with its members (from $\mathcal{R}_k^{\text{intra}}$) and $n_k^{\text{inter}}$ inter-unit relations (from $\mathcal{R}^{\text{inter}}$). Therefore, the block-smoothness constant of the anchor pose scales as $(n_k^{\text{intra}} + n_k^{\text{inter}}) \cdot \beta$. We show that this linear scaling is attainable via a star graph example, which is a frequently observed substructure in practice.

**Tightness via a star graph.** Consider a star constraint graph with a root $x_0$ and $M$ leaf nodes $\{x_1, \ldots, x_M\}$, as well as quadratic penalties $\mathcal{L}_\star = \frac{1}{2} \sum_{i=1}^M \|(x_i - x_0) - d_i\|^2$, where $d_i$ are target distance between the root and the leaf nodes. Replacing $x_0$ by $x_0 + \delta$ shifts $\nabla \mathcal{L}_\star$ by $\sum_{i=1}^M \delta = M\delta$. The smoothness bound $n\beta$ is thus achieved with equality.

Intuitively, this illustrates how updating an anchor to satisfy one relation simultaneously perturbs all its other incident constraints, which can lead to oscillatory updates and slow convergence.

## B.2. Re-parameterization Removes Intra-Unit Topological Coupling

A key property of intra-unit relations is that they depend only on relative poses among members within the same unit, not on the global placement of the entire unit. The global-to-local re-parameterization exploits this invariance.

Formally, for a unit $U_k$ with global pose $P_k$ and member local poses $p_i^\ell$, the member global pose is $p_i = P_k \oplus p_i^\ell$ (Eq. (2)). Thus, for any two members $a_i, a_j \in U_k$, their relative transformation satisfies the standard cancellation identity:

$$\underbrace{(P_k \oplus p_i^\ell)^{-1} \oplus (P_k \oplus p_j^\ell)}_{i \to j \text{ in global frame}} = \underbrace{(p_i^\ell)^{-1} \oplus p_j^\ell}_{i \to j \text{ in unit frame}} \qquad (21)$$

**Proposition B.1** (Intra-unit gradient cancellation). *Let*

$$\ell(r; \tilde{p}) = \varphi\big(p_i^{-1} \oplus p_j\big) \qquad (22)$$

*be the differentiable intra-unit penalty between members $a_i, a_j \in U_k$ of the same unit, where $r \in \mathcal{R}_k^{intra}$ is their intra-unit relation, and $\varphi$ is an arbitrary differentiable function. Then, under the global-to-local re-parameterization*

$$p_i = P_k \oplus p_i^\ell, \quad p_j = P_k \oplus p_j^\ell \qquad (23)$$

*we have*

$$\nabla_{P_k} \ell(r; \tilde{p}) = 0 \qquad (24)$$

*Proof.* Substituting re-parameterized poses into the penalty:

$$
\begin{aligned}
\ell(r; \tilde{p}) &= \varphi(p_i^{-1} \oplus p_j) \\
&= \varphi\big((P_k \oplus p_i^\ell)^{-1} \oplus (P_k \oplus p_j^\ell)\big) \\
&= \varphi\big((p_i^\ell)^{-1} \oplus P_k^{-1} \oplus P_k \oplus p_j^\ell\big) \\
&= \varphi\big((p_i^\ell)^{-1} \oplus p_j^\ell\big)
\end{aligned}
\tag{25}
$$

Since $\ell(r; \tilde{p})$ is a function of the relative poses $p_i^\ell$ and $p_j^\ell$ alone, without dependence on $P_k$, $\nabla_{P_k}\ell(r; \tilde{p}) = 0$. □

**Consequence.** Under global parameterization, intra-unit penalties incident to a unit anchor contribute to the anchor's gradient, creating an $O(n_k^{\text{intra}})$ stiffness accumulation. Under our global-to-local re-parameterization, on the other hand, all intra-unit penalties satisfying Proposition B.1 contribute zero gradient to the unit pose $P_k$. The block-smoothness bound for $P_k$ is therefore reduced from $(n_k^{\text{intra}} + n_k^{\text{inter}}) \cdot \beta$ down to $n_k^{\text{inter}} \cdot \beta$. The intra-unit relation induced stiffness accumulation is removed by construction.

### B.3. Discussion

Proposition B.1 provides a structural guarantee: global-to-local re-parameterization eliminates a specific gradient coupling whose magnitude scales linearly with the number of intra-unit relations $n_k^{\text{intra}}$. This does not provide convergence rate guarantee, as the overall optimization landscape also depends on a multitude of things, such as the loss geometry, inter-unit interactions, collision and boundary terms, etc.

Nevertheless, removing an $O(n_k^{\text{intra}})$ accumulation source from each unit is strictly preferable to leaving it in place. The re-parameterization also introduces no additional hyper-parameter, and we observe strong empirical gains as shown by the convergence curves in Figure 6 and Figure 16.

## C. Details of Invariant Spatial Decomposition

We implement invariant spatial decomposition by employing the prompt detailed below. Specifically, we enforce this decomposition **by construction** by mandating that each unit contains exactly one anchor and that unit members are strictly prohibited from having inter-unit relations. Any violation of these constraints triggers a failure in our DSL syntactic checker, necessitating a retry in a subsequent API session. Empirically, such failure modes are rare, with nearly all tasks successfully completed in a single pass.

```
Complex scenes are often composed of
↪   multiple unique frame of references,
↪   that are prone to
↪   orientation-related errors. You
↪   MUST use units to break the problem
↪   into two distinct stages to avoid
↪   mental rotation:
```

```
1. STAGE 1: LOCAL ASSEMBLY (Inside a
↪   unit):
    - Ignore the room. Ignore walls.
    ↪   Ignore other units.
    - Imagine the Anchor asset
    ↪   effectively becomes the origin
    ↪   of a small, local universe.
    - The Anchor is fixed at (0,0) and
    ↪   locked facing +Y in this local
    ↪   frame.
    - Assemble member objects relative
    ↪   *only* to the Anchor.
    - This creates a rigid
    ↪   "pre-assembled unit."

2. STAGE 2: GLOBAL PLACEMENT (Outside
↪   units):
    - Once a unit is formed, ignore the
    ↪   unit internals. Forget about its
    ↪   members.
    - You can now manipulate the Unit
    ↪   Handle as a **single rigid
    ↪   entity**, just like independent
    ↪   Assets.
    - You place that entire unit into
    ↪   the room by applying constraints
    ↪   to the unit handle w.r.t to
    ↪   other handles / independent
    ↪   Assets.

RULES:
- **Unit Formation**:
    - Units are spatial reasoning tools
    ↪   aimed at reducing mental
    ↪   rotation and problem
    ↪   complexity.
    - Create units when you find
    ↪   reasoning an object's
    ↪   orientation difficult.
    - On the constraint graph, this
    ↪   usually means factorizing a
    ↪   star-shaped component into a
    ↪   unit.
- **Internal Scope:** Inside the `with`
↪   block, you may ONLY reference the
↪   `members` list.
    - FORBIDDEN Internally:
    ↪   `against_wall`, `corner`,
    ↪   `horizontal`, `vertical`, or
    ↪   `facing(..., target=<WallId>)`.
- **Global Scope:** Outside the block,
↪   you may ONLY reference independent
↪   Assets and UnitHandles.
    - FORBIDDEN Globally: Referencing a
    ↪   unit member asset individually
    ↪   (except via the handle).
- **Var Variables:** Declared at the
↪   TOP level. They pass through
↪   scopes.

DSL USAGE:
```

```
In terms of DSL, think of unit members
↪   as "private": they are only
↪   referenced inside the unit `with`
↪   block.
Once a unit is formed, you can forget
↪   about its internals, and manipulate
↪   it as a single rigid object.
Outside the block, you must use the
↪   unit handle.
No nested units. No shared members
↪   between units. No cross-scope
↪   references.

--------------------------

DSL SPECIFICATION

--------------------------

```python
class ConstraintSolver:
  def __init__(self):
      self.constraints = []
  @contextmanager
  def unit(self, unit_id: str, anchor:
  ↪   Asset, members: List[Asset]) ->
  ↪   Iterator[UnitHandle]:
      """
      Declare a layout unit.

      - Use as a context manager.
      - All solver.* calls inside the
      ↪   `with` block are recorded as
      ↪   INTERNAL constraints for this
      ↪   unit.
      - The yielded UnitHandle is used
      ↪   ONLY in GLOBAL constraints
      ↪   outside the block.

      Args:
      - unit_id: unique id string for
      ↪   this unit
      - anchor: anchor Asset (defines
      ↪   the unit's global pose)
      - members: list of member Assets
      ↪   (must include anchor; no
      ↪   duplicates)
      """
      assert anchor in members
      yield UnitHandle(unit_id=unit_id)
```
```

## D. Details of Consistent Spatial Imagination

We employ the following prompt to implement consistent spatial imagination. Specifically, by mandating that the MLLM output the bounding box, oriented footprint, and spatial extent (*i.e.*, the $X$ and $Y$ bounds occupied by a unit) in a DSL format, we elicit an explicit externalization of its internal mental model. This process compels the MLLM to concretely reason about spatial layouts rather than relying on vague associations. Any omission of these geometric primitives triggers a failure in our DSL syntactic checker, resulting in an automated retry via a new API session. Empirically, such failures are rare, and nearly all tasks are successfully completed in a single pass.

```
You are required to output your spatial
↪   mental model to ensure geometric
↪   consistency.

**SPATIAL SCAFFOLD: 3-STEP PATTERN
↪   (MANDATORY)**

For EACH object/unit, explicitly
↪   compute in sequence:
```python
<obj>.footprint = Footprint(rz=<deg>,
↪   lx=<float>, ly=<float>)  #
↪   dimensions after rotation
<obj>.pose = Pose(x=<float>, y=<float>,
↪   rz=<deg>)              # placement
↪   of rotated footprint
<obj>.bounds = Bounds(x_min=<float>,
↪   x_max=<float>, ...)       #
↪   estimate spans for collision
↪   detection
```

**Rotation → Dimension Table:**
| rz (degrees) | Footprint (lx, ly) |
↪   Rule |
| 0, 180       | (sx, sy)           | NO
↪   SWAP |
| 90, -90      | (sy, sx)           |
↪   SWAP x<->y |
| other        | \|sx·cos(rz)\| +
↪   \|sy·sin(rz)\|, \|sx·sin(rz)\| +
↪   \|sy·cos(rz)\| | General formula |

**Bounds Derivation:** `x_min, x_max =
↪   cx - lx/2, cx + lx/2` (same for y)
**Collision Check:** Overlap in BOTH x
↪   AND y ranges = collision

You must explicitly calculate the
↪   geometry in two isolated stages to
↪   avoid mental rotation errors.

**THE GOLDEN RULE**
Do NOT attempt to visualize the whole
↪   room at once. Build locally, seal,
↪   then place globally.

* STAGE 1: THE LOCAL VOID (Inside `with
↪   solver.unit...`)
    - Infinite Void: Inside a unit
    ↪   block, the room and walls cease
    ↪   to exist. You are in an
    ↪   infinite void.
    - Anchor is Origin: The anchor is
    ↪   strictly fixed at `(0, 0)`
    ↪   facing `+Y` (`rz=0`).
```

```
        - Relative Estimates: estimate pose
        ↪  and footprint for all members
        ↪  relative *only* to this
        ↪  unrotated anchor.
            - Benefit: You reason purely in
            ↪  standard alignment
            ↪  (front/back/left/right)
            ↪  without complex mental
            ↪  rotation.

  * STAGE 1.5: SEAL THE BLACKBOX
        - Final Check: For wall-bound units,
        ↪  no objects extend behind the
        ↪  anchor; rear edge is flush.
        - Compute AABB: Based on these
        ↪  local poses, calculate the
        ↪  unit's bounding box (lx, ly).
        - Sealed: The unit is now a rigid
        ↪  unit with defined dimensions
        ↪  and mounting orientation.

  * STAGE 2: GLOBAL MAPPING (Outside
  ↪  units)
        - Room Exists: Now, the room exists.
        ↪  You are placing "Black Boxes"
        ↪  (UnitHandles) and independent
        ↪  Assets.

        - Block Placement: Determine the
        ↪  global pose and footprint of
        ↪  the UnitHandles and independent
        ↪  Assets using room constraints
        ↪  (walls, corners).

        - No Peeking: Do not calculate
        ↪  global poses for individual
        ↪  unit anchors or members. The
        ↪  Handle points to the AABB's
        ↪  center.

--------------------------

DSL SPECIFICATION

--------------------------

```python
class Pose(BaseModel):
  x: float = Field(description="X
  ↪  coordinates of the entity's
  ↪  center")
  y: float = Field(description="Y
  ↪  coordinates of the entity's
  ↪  center")
  rz: float =
  ↪  Field(description="Rotation in
  ↪  degrees. Positive value means
  ↪  counter-clockwise from +Y")

class AABB(BaseModel):
  """
    Axis-Aligned Bounding Box (in unit
    ↪  Local Frame).
    Used to estimate the footprint of a
    ↪  unit, for global pose estimation
    """
    lx: float = Field(description="Total
    ↪  Length along X axis (width)")
    ly: float = Field(description="Total
    ↪  Length along Y axis (height)")

class Footprint(BaseModel):
  """
    Rotated bounding box dimensions
    ↪  (reasoning scaffold).

    Rotation rules (for original size sx,
    ↪  sy):
        rz in {0, 180}: lx=sx, ly=sy (NO
        ↪  SWAP)
        rz in {90, -90}: lx=sy, ly=sx
        ↪  (SWAP)
        other angles: lx = |sx*cos(rz)| +
        ↪  |sy*sin(rz)|
                      ly = |sx*sin(rz)| +
                      ↪  |sy*cos(rz)|
    """
    rz: float =
    ↪  Field(description="Rotation angle
    ↪  (degrees). Must match intended
    ↪  pose.rz.")
    lx: float =
    ↪  Field(description="Footprint
    ↪  length along X-axis AFTER
    ↪  rotation.")
    ly: float =
    ↪  Field(description="Footprint
    ↪  length along Y-axis AFTER
    ↪  rotation.")

class Bounds(BaseModel):
  """
    Axis-aligned bounding box occupancy
    ↪  (collision scaffold).

    Derivation: Given pose center (cx,
    ↪  cy) and footprint (lx, ly):
        x_min, x_max = cx - lx/2, cx +
        ↪  lx/2
        y_min, y_max = cy - ly/2, cy +
        ↪  ly/2

    Collision: Two objects collide iff
    ↪  bounds overlap in BOTH x AND y.
    """
    x_min: float =
    ↪  Field(description="Minimum X
    ↪  coordinate of bounding box.")
    x_max: float =
    ↪  Field(description="Maximum X
    ↪  coordinate of bounding box.")
```

```
    y_min: float =
    ↪    Field(description="Minimum Y
    ↪    coordinate of bounding box.")
    y_max: float =
    ↪    Field(description="Maximum Y
    ↪    coordinate of bounding box.")

class UnitHandle(BaseModel):
    """
    Opaque handle representing a unit in
    ↪    GLOBAL scope.
    """
    unit_id: str =
    ↪    Field(description="Unique id for
    ↪    the unit")

    aabb: Optional[AABB] =
    ↪    Field(default=None,
    ↪    description="Estimated Local
    ↪    Bounding Box of the unit")
    pose: Optional[Pose] =
    ↪    Field(default=None,
    ↪    description="Global pose of the
    ↪    Anchor asset")

class Asset(BaseModel):
    id: str = Field(description="Unique
    ↪    id for the asset")
    description: str =
    ↪    Field(description="Description of
    ↪    the asset")
    size: Tuple[float, float] =
    ↪    Field(description="2D footprint
    ↪    size of the asset (x, y) in
    ↪    meters")

    pose: Optional[Pose] =
    ↪    Field(default=None,
    ↪    description="Estimated pose.
    ↪    Local if in unit, Global if
    ↪    independent.")
```

# E. Details of Supportive Spatial Optimization

## E.1. Learnable Parameter Mechanism

Our supportive spatial optimization incorporates a **parameter-sharing mechanism**. For relational instances anticipated to exhibit the same metric scale (*e.g.*, symmetric counterparts such as the two chair–table distances in a dining set), we bind them to a single underlying metric parameter during the optimization process. This is particularly effective when collision losses repel objects: parameter sharing prevents symmetric relation metrics from drifting independently after overlaps are resolved, thereby preserving consistent relational distances across symmetric instances.

Furthermore, we utilize a **two-stage optimization strategy**. In the first stage, we optimize object poses with all constraint parameters fixed and physical constraints (*e.g.*,

collision loss) disabled. This allows the optimizer to first attain a stable configuration that fulfills all semantic requirements. In the second stage, we enable learnable constraint parameters and physics constraints, allowing the system to *finetune* the layout from its previous state and resolve physical violations via the shared parameters.

Benefiting from this parameter sharing, adjustments to constraint parameters typically do not result in a significant degradation of layout quality. For example, in a grid of tables where each row shares a uniform vertical position parameter, updating the parameter affects the *entire* row simultaneously, preventing individual objects from losing their alignment. To prevent collapse into trivial layouts (*i.e.*, the optimizer setting parameter values that minimize loss but violate semantic constraints), we add a *prior loss* to regularize the parameters and inhibit arbitrary drift.

## E.2. Optimizer Details

*Table 8.* Fixed set of hyperparameters used by all scenes.

| Parameter | Value |
|---|---|
| Optimizer | SGD (momentum = 0.9) |
| Iterations | 600 |
| Learning rate (position) | 0.5 |
| Learning rate (rotation) | 0.3 |
| Scheduler | Cosine annealing |
| Gradient clipping (position) | 1.0 |
| Gradient clipping (rotation) | 0.3 |

The optimizer operates over three fundamental tensor representations: (i) the **pose vector** $\mathbf{p} = (\mathbf{x}, \mathbf{y}, \boldsymbol{\theta}) \in \mathbb{R}^{3 \times N}$, where $\mathbf{x}_i, \mathbf{y}_i$ denotes the 2D position of object $i$, and $\boldsymbol{\theta}_i$ denotes its rotation (in radians); (ii) the **bounding box vector** $\mathbf{b} = (w_x, w_y) \in \mathbb{R}^{2 \times N}$, where $w_x, w_y$ are half-widths along the local axes; and (iii) the **parameter vector** $\phi \in \mathbb{R}^P$ (as described in E.1).

The bounding box vector $\mathbf{b}$ is fixed and provided by the asset library, while the parameter vector $\phi$ is determined by the MLLM. The pose vector $\mathbf{p}$ is randomly initialized at the start of optimization.

We employ a standard Cosine Annealing learning rate schedule with the minimum learning rate $\eta_{min}$ set to 0. Since our pose parameters span several orders of magnitude (*e.g.*, from 10 m → 1 cm), this annealing schedule is necessary to prevent saturation. We use a standard Stochastic Gradient Descent optimizer with a momentum of 0.9. Separate learning rates are applied to position and rotation, and gradient clipping is used to suppress gradient explosions, particularly when random initialization yields a suboptimal starting $\mathbf{p}$.

In practice, on a single RTX 4090 GPU, a single optimization stage (600 iterations) completes in 15 seconds. Con-

sequently, the total runtime for our dual-stage optimization process is roughly 30 seconds per scene. All qualitative and quantitative results presented in this paper are generated using the fixed hyperparameter set detailed in Table 8.

# F. Differentiable Spatial Constraints

We define differentiable objectives for the spatial constraints used in Supportive Spatial Optimization. Following the main text, the pose of asset $a_i$ is $p_i = (x_i, y_i, \theta_i)$, its planar bounding box size is $(l_i, w_i)$, and the room has planar dimensions $(L, W)$. The orientation vector is $\mathbf{v}_i = (\cos\theta_i, \sin\theta_i)$. For unit $U_k$, the unit pose is $P_k = (x_k, y_k, \theta_k)$, and member assets have local poses $p_i^\ell$. We denote $\phi(z) = \max(0, z)^2$ as the soft penalty, $R(\theta)$ for the 2D rotation matrix.

**Overall Objective.** We optimize the mixed representation $\tilde{p}$:

$$\mathcal{L}(\tilde{p}) = \mathcal{L}_{\text{global}}(\tilde{p}) + \sum_{k=1}^{K} \mathcal{L}_{\text{local}}^k(\tilde{p}), \qquad (26)$$

where global and local terms include collision, boundary, and relational losses with weights $\lambda_{\text{col}}, \lambda_{\text{bd}}, \lambda_{\text{rel}}$.

**Collision Loss.**

$$\mathcal{L}_{\text{col}} = \sum_{i<j} \left[ \text{IoU}_{ij} - \frac{d_{ij}^2}{c^2} \rho_{ij} \right], \qquad (27)$$

where $d_{ij}$ is the center distance, $c$ the diagonal of the smallest enclosing box, and $\rho_{ij} = |B_i \cap B_j| / \min(|B_i|, |B_j|)$ the overlap ratio that smoothly modulates the distance penalty.

Notably, we added a gating term $\rho_{ij}$. In traditional DIoU, the distance penalty would repel *all* box pairs regardless of collision, producing overly sparse layouts in scenes where tight packing is desired. By activating the penalty only in proportion to the actual overlap, the loss resolves collisions while preserving spatial compactness when needed.

**Boundary Loss.**

$$\mathcal{L}_{\text{bd}} = \sum_{i} \sum_{k=1}^{4} \|c_{i,k} - \text{clamp}(c_{i,k}, \mathbf{0}, \mathbf{u})\|_1 \qquad (28)$$

where $c_{i,k} \in \mathbb{R}^2$ are the four corners of asset $a_i$ after rotation and $\mathbf{u} = (L, W)^\top$ is the room boundary ($L$ is length and $W$ is width). Intuitively, this is the sum $L_1$ distances by which the four corners of the asset's OBB exceed the room boundary.

**Distance Loss.**

$$\mathcal{L}_{\text{dist}}(p_i, p_j, d^*) = (\|(x_i, y_i) - (x_j, y_j)\|_2 - d^*)^2. \quad (29)$$

where $d^*$ is the targeted center-to-center distance between assets $i$ and $j$ that we want to achieve.

**Gap Loss.**

$$\mathcal{L}_{\text{gap}}(B_i, B_j, g) = \left(D_{\min}(B_i, B_j) - g\right)^2, \qquad (30)$$

where $B_i = (p_i, s_i, \theta_i)$ denotes an OBB with center $p_i \in \mathbb{R}^2$, half-extents $s_i \in \mathbb{R}^2$, and orientation $\theta_i$, and $D_{\min}$ is the minimum boundary distance approximated via vertex-to-OBB signed distance functions. This enforces a minimum clearance $g$ between asset $a_i$ and $a_j$'s bounding boxes.

**Against-Wall Loss.**

$$\mathcal{L}_{\text{wall}}(p_i, w) = (p_i^{(w)} - t_w)^2 \\ + 1 - \cos(\theta_i - \theta_w^*)), \qquad (31)$$

where $p_i^{(w)} \in \{x_i, y_i\}$ is the coordinate constrained by wall $w$, $t_w$ is the target position accounting for object extent, $\theta_w^*$ is the orientation perpendicular to $w$. This loss positions asset $i$ against wall $w \in \{\text{L}, \text{R}, \text{T}, \text{B}\}$.

**Corner Loss.**

$$\mathcal{L}_{\text{corner}}(p_i, c, w) = (x_i - x_c^*)^2 + (y_i - y_c^*)^2 \\ + 1 - \cos(\theta_i - \theta_w^*), \qquad (32)$$

where $c \in \{\text{BL}, \text{BR}, \text{TR}, \text{TL}\}$ specifies the corner and $w$ selects one of its two adjacent walls to place the object against, resolving placement ambiguity. The target position $(x_c^*, y_c^*)$ offsets from corner $c$ toward the room interior by the object's half-extents, with the offset direction determined by $w$. The target orientation $\theta_w^*$ faces perpendicular to wall $w$ into the room. Intuitively, this loss snaps an object into a corner, flush against a designated wall $w$ and facing inward.

**Facing Constraint.**

$$\mathcal{L}_{\text{face}}(p_i, p_j) = 1 - \frac{\mathbf{v}_i \cdot \mathbf{d}_{ij}}{\|\mathbf{d}_{ij}\|}, \qquad (33)$$

where $\mathbf{v}_i = (\cos\theta_i, \sin\theta_i)$ is the unit orientation vector of asset $a_i$, and $\mathbf{d}_{ij} = (x_j - x_i, y_j - y_i)$ is the displacement from $a_i$ to $a_j$. This rotates asset $a_i$ to face target $a_j$. In practice, it is possible that $\|\mathbf{d}_{ij}\| = 0$; we address this by introducing a small $\epsilon$ constant.

**Directional Constraints.** For relative positioning (left-of, right-of, in-front-of, behind-of), we transform source $s$ into target $t$'s local frame:

$$(x', y') = R(\theta_t)^\top (p_s - p_t). \qquad (34)$$

Each direction applies two terms—a *directional* term $\mathcal{L}_{\text{dir}}$ and an *alignment* term $\mathcal{L}_{\text{align}}$:

$$\mathcal{L}_{\text{left}} = \underbrace{\phi(x' + r_x + e_x)}_{\mathcal{L}_{\text{dir}}} + \underbrace{|y' - \bar{y}|}_{\mathcal{L}_{\text{align}}}, \qquad (35)$$

$$\mathcal{L}_{\text{right}} = \phi(e_x - x' + r_x) + |y' - \bar{y}|, \qquad (36)$$

$$\mathcal{L}_{\text{front}} = \phi(e_y - y' + r_y) + |x' - \bar{x}|, \qquad (37)$$

$$\mathcal{L}_{\text{behind}} = \phi(y' + r_y + e_y) + |x' - \bar{x}|, \qquad (38)$$

where $(x', y')$ is the source center in the target's local frame, $R(\theta_t)$ is the 2D rotation matrix, and $p_s, p_t \in \mathbb{R}^2$ are global positions. The terms $e_x, e_y$ are the target's half-extents, and $r_x, r_y$ are the source's half-extents projected onto the target's local axes. The function $\phi(z) = \max(0, z)$ penalizes violations. The alignment targets are $\bar{y} = (2p-1)(e_y - r_y)$ and $\bar{x} = (2p-1)(e_x - r_x)$ with $p \in [0, 1]$.

Intuitively, $\mathcal{L}_{\text{dir}}$ enforces that the source's bounding box lies entirely on the specified side of the target (e.g., left or in-front), while $\mathcal{L}_{\text{align}}$ controls where along the perpendicular axis the source is positioned by smoothly interpolating between edge-aligned placements: $p=0$ aligns the source with one edge of the target, $p=1$ with the opposite edge, and $p=0.5$ yields strict centering.

**Angle Loss**

$$\mathcal{L}_{\text{angle}}(p_i, p_j, \alpha) = 1 - \cos(\theta_i - \theta_j - \alpha), \qquad (39)$$

where $\theta_i, \theta_j$ are the yaw angles of assets $a_i, a_j$. This loss enforces a fixed angular offset $\alpha$ between the two assets.

**Placement Loss.**

$$\mathcal{L}_{\text{h-place}}(p_i, x^*) = \phi\big(|x_i - x^*| - mL\big), \qquad (40)$$

$$\mathcal{L}_{\text{v-place}}(p_i, y^*) = \phi\big(|y_i - y^*| - mW\big), \qquad (41)$$

where $\phi(z) = \max(0, z)^2$ is a soft penalty that activates when the violation exceeds a threshold, $(x^*, y^*)$ is the target position for asset $a_i$, $(L, W)$ are room dimensions, and $m$ is a margin factor allowing slight deviation from the target. In practice, we set $m$ to zero.

**Around Arrangement.** For arranging $N$ assets around a focal point with sweep angle $S$:

$$\mathcal{L}_{\text{around}} = \frac{1}{N-1} \sum_{i=1}^{N-1} \left(\Delta_i - \frac{S}{N-1}\right)^2 \qquad (42)$$
$$+ \|\bar{\mathbf{u}} - \bar{\mathbf{u}}^*\|^2,$$

where $\Delta_i$ are angular gaps between sorted positions, $\bar{\mathbf{u}} = \frac{1}{N} \sum_i (\sin\theta_i, \cos\theta_i)$ is the circular mean embedding of source angles in the target's local frame, and $\bar{\mathbf{u}}^* = M(\sin c, \cos c)$ with $M = \sin(N\delta)/(N\sin\delta)$ for $\delta = S/[2(N-1)]$ is the target embedding centered at $c$.

# G. Details of Evaluation

### G.1. Details of Test Case Generation

We follow the test case generation protocol of (Sun et al., 2025), but do not directly reuse its benchmark because it is not fully aligned with our setting. Specifically, existing benchmarks often use short, simple instructions, which are insufficient for evaluating compositional relative spatial reasoning. In addition, our framework focuses on floor objects, while prior benchmarks include various small tabletop or non-floor objects that fall outside our scope.

To address these discrepancies while ensuring a fair comparison, we regenerate and filter the test cases to better match our setting while keeping the evaluation protocol aligned with prior work. Concretely, for each of the nine scene types, we construct three scenes with predefined room sizes. We then utilize GPT-5 to generate instructions across three difficulty levels, accompanied by plausible asset lists. The corresponding 3D assets are retrieved from Objaverse (Deitke et al., 2023). Finally, we performed human verification to remove or replace low-quality assets, such as those with mislabeled categories or low-poly geometry. We employ the following prompt for test case generation.

```
You are a professional interior
↪   designer who has designed thousands
↪   of functional and aesthetically
↪   pleasing interiors.

Given a room type, room size (length m
↪   x width m), and difficulty level
↪   (easy / medium / hard), generate a
↪   practical and realistic layout
↪   instruction for the room.

Only include key floor-standing
↪   furniture or other large floor
↪   objects. Do not include small items,
↪   wall-mounted objects, ceiling
↪   objects, or tabletop decorations.

Difficulty levels:

- easy: few objects, simple
↪   compositional spatial relations
- medium: moderate number of objects,
↪   more compositional spatial
↪   relations
- hard: more objects, complex
↪   multi-step spatial relations and
↪   stronger global coordination

Requirements:
1. First provide a concise high-level
↪   description of the overall layout
↪   and design strategy.
2. Then list the key objects. For each
↪   object, provide:

- description: a short
↪   retrieval-friendly description
- size: [length, width, height] in
↪   centimeters
- quantity: integer
- variance_type: "same" or "varied"
```

```
3. Write one coherent paragraph as the
↪  instruction. It should describe the
↪  overall layout strategy, the listed
↪  objects, and the spatial
↪  arrangement among these objects
↪  using clear relative relations such
↪  as left of, right of, in front of,
↪  behind, next to, facing, aligned
↪  with, against a wall, near a corner,
↪  or centered in.

4. The layout must be feasible,
↪  functional, and consistent with the
↪  given room type, size, and
↪  difficulty level.

Output valid JSON only:
{
  "difficulty": "easy | medium | hard",
  "instruction": "...",
  "objects": {
    "object_name": {
      "description": "...",
      "size": [100, 60, 75],
      "quantity": 1,
      "variance_type": "same"
    }
  }
}
```

### G.2. Details of Physical Evaluation

For physical evaluation, we report the collision ratio (%CR) and out-of-bound ratio (%OR), defined as the percentage of objects involved in collision or penetrating wall boundaries.

**Collision Ratio (%CR).** Given a scene with $n$ objects represented as 2D bounding polygons $\{P_1, P_2, \ldots, P_n\}$ projected onto the floor plane, we compute pairwise intersections between all $\binom{n}{2}$ object pairs. An object pair $(P_i, P_j)$ is considered *colliding* if their intersection area exceeds a tolerance threshold $\tau_c$:

$$\text{collide}(P_i, P_j) = \mathbf{1}\left[\text{area}(P_i \cap P_j) > \tau_c\right] \quad (43)$$

We then count the number of unique objects involved in at least one collision:

$$\mathcal{C} = \{i \mid \exists j \neq i : \text{collide}(P_i, P_j) = 1\} \quad (44)$$

The collision ratio is defined as:

$$\%\text{CR} = \frac{|\mathcal{C}|}{n} \times 100 \quad (45)$$

**Out-of-Bound Ratio (%OR).** Given the room boundary polygon $R$ (derived from the wall vertices), an object $P_i$ is considered *out-of-bound* if it extends beyond the room boundary by more than a tolerance threshold $\tau_o$:

$$\text{oob}(P_i) = \mathbf{1}\left[\text{area}(P_i) - \text{area}(P_i \cap R) > \tau_o\right] \quad (46)$$

The out-of-bound ratio is defined as:

$$\%\text{OR} = \frac{\sum_{i=1}^{n} \text{oob}(P_i)}{n} \times 100 \quad (47)$$

**Tolerance Margins.** We employ tolerance margins $\tau_c = 0.0003\,\text{m}^2$ and $\tau_o = 0.0003\,\text{m}^2$ to prevent false positives arising from numerical uncertainty in gradient-based optimization. Methods including LayoutVLM and R³L use continuous optimization to refine object placements, which can introduce floating-point imprecision at object boundaries. Without these margins, negligible sub-centimeter overlaps (on the order of $10^{-6}\,\text{m}^2$) would be incorrectly flagged as collisions, unfairly penalizing optimization-based methods compared to DFS-based approaches that place objects on discrete grid (*i.e.*, Holodeck) and approaches that output precise coordinates (*i.e.*, LayoutGPT). Since our goal is to assess relative spatial reasoning, neglecting physically insignificant overlaps is a sensible decision.

**Implementation Details.** Object bounding polygons are computed from the 3D asset bounding boxes projected onto the 2D floor plane, accounting for object rotation. The room polygons are derived from the convex hull of the end points of the wall segments. All geometric operations (intersection, area computation) are performed using the Shapely library with double-precision floating-point arithmetic.

### G.3. Details of Absolute Semantic Evaluation

We employ the following prompt to query the MLLM and generate the absolute semantic scores (*i.e.*, Real., Func. and Instr.) presented in Table 1.

```
You are an expert evaluator for 3D
↪  scene layout generation.

You are given:
- a top-down view image of the
↪  generated scene,
- a side view image of the generated
↪  scene,
- the text instruction provided to the
↪  3D layout generator.

Your task is to evaluate the generated
↪  scene using the criteria below.
Assign **integer scores from 1 to 10**
↪  and provide a brief justification.

Focus only on semantic, spatial, and
↪  functional layout aspects. Ignore
↪  texture, material, lighting, object
↪  count, and visual style.
Pay attention to whether each object's
↪  orientation is reasonable (e.g.,
↪  whether the interactive face faces
↪  free space rather than a wall).
```

```
**Score Scale (integers only, no
↪   decimals)**
- 10: Flawless | exceeds expectations,
↪   no issues
- 9: Excellent | negligible issues only
- 8: Very good | minor cosmetic issues
- 7: Good | minor issues that don't
↪   affect function
- 6: Acceptable | noticeable but
↪   limited issues
- 5: Mediocre | issues that somewhat
↪   reduce quality
- 4: Below average | clear issues
↪   affecting usability
- 3: Poor | significant issues
↪   hindering intended use
- 2: Very poor | severe issues, barely
↪   functional
- 1: Unusable | fails to meet basic
↪   expectations

**Evaluation Criteria**

1. **Realism**:
   How believable the layout is given
   ↪   common-sense physical and
   ↪   spatial expectations:
   ↪   collision-free placement, no
   ↪   out-of-bounds objects, and all
   ↪   functional faces oriented toward
   ↪   the room interior.
   - 8-10: Believable, logically
   ↪   arranged, no obvious
   ↪   implausibilities
   - 4-7: Generally plausible, but
   ↪   noticeable implausibilities
   ↪   (awkward spacing, unnatural
   ↪   orientation)
   - 1-3: Clearly implausible (extreme
   ↪   crowding, impossible placement,
   ↪   highly unnatural arrangement)

2. **Functionality**:
   How well the layout supports
   ↪   functional use:
   ↪   access/interaction space, object
   ↪   affordances via
   ↪   placement/orientation, and
   ↪   functional zoning.
   - 8-10: Intended use well supported;
   ↪   access and interaction space
   ↪   sufficient; zones match purposes
   - 4-7: Partially usable; some access
   ↪   constrained; noticeable blockers
   ↪   reduce usability
   - 1-3: Largely unusable; major
   ↪   access blocked; key objects
   ↪   cannot be used

3. **Instruction Following**:
   Whether the layout satisfies the
   ↪   semantic spatial relationships
   ↪   described in the instruction.
```

```
   Do not penalize unrealism if it is
   ↪   faithful to the instruction.
   - 8-10: Key instructed spatial
   ↪   relationships satisfied; at most
   ↪   minor mismatches
   - 4-7: Some key relationships
   ↪   violated or ambiguous;
   ↪   instruction only partially
   ↪   reflected
   - 1-3: Most key relationships
   ↪   violated or contradicted; layout
   ↪   fails to reflect instruction

---

The text instruction is:
```${instruction}```

---

Output JSON only (no extra text).
↪   Scores must be integers 1-10:
```json
{
    "realism": {
      "score": <integer 1-10>,
      "justification": "..."
    },
    "functionality": {
      "score": <integer 1-10>,
      "justification": "..."
    },
    "instruction_following": {
      "score": <integer 1-10>,
      "justification": "..."
    }
}
```
```

For each task × baseline combination, we query Gemini 3 Flash five times and average the resulting scores to obtain the final values. During each evaluation pass, the model is provided with both the top-down and side-view renderings of the generated scene.

### G.4. Details of Pairwise Semantic Evaluation

We employ the following prompt to query the MLLM to select a preferred layout between two methods, given their outputs on the same task.

```
You are an expert judge comparing two
↪   3D scene layouts.

You are given a composite image with:
- Layout A: LEFT column (top-down view
↪   above, side view below)
- Layout B: RIGHT column (top-down view
↪   above, side view below)

Both layouts were generated from the
↪   same instruction:
```
```

```
${instruction}
```

Focus only on semantic, spatial, and
↪  functional layout aspects. Ignore
↪  texture, material, lighting, object
↪  count, and visual style.
Pay attention to whether each object's
↪  orientation is reasonable (e.g.,
↪  whether the interactive face faces
↪  free space rather than a wall).

Compare the layouts on these criteria:

1. **Spatial Realism**: How believable
↪  the layout is given common-sense
↪  physical and spatial expectations:
↪  collision-free placement, no
↪  out-of-bounds objects, and all
↪  functional faces oriented toward
↪  the room interior.

2. **Functionality**: How well the
↪  layout supports functional use:
↪  access/interaction space, object
↪  affordances via
↪  placement/orientation, and
↪  functional zoning.

3. **Instruction Following**: Whether
↪  the layout satisfies the semantic
↪  spatial relationships described in
↪  the instruction. Do not penalize
↪  unrealism if it is faithful to the
↪  instruction.

Choose the layout that is better
↪  overall. You MUST pick a winner.

Output JSON only:
```json
{"winner": "A"}
```
or
```json
{"winner": "B"}
```

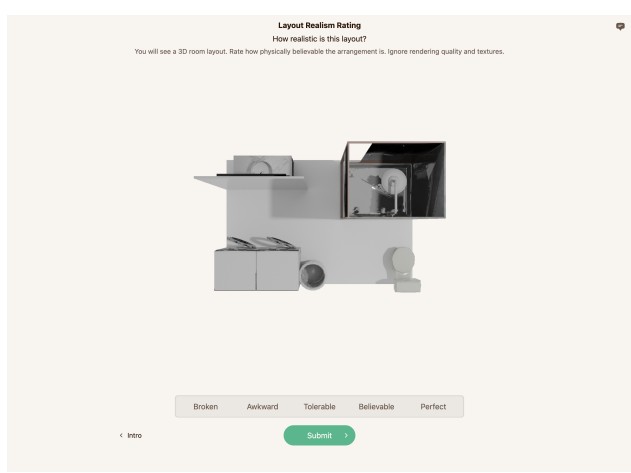

*Figure 7.* User study interface for absolute semantic rating.

This transformation corresponds a 400-point Elo difference to approximately $10:1$ odds (*i.e.*, $P(m_i \succ m_j) \approx 0.91$ when $\text{Elo}_i - \text{Elo}_j = 400$), consistent with the standard Elo rating system convention.

We fit Bradley-Terry on per-task outcome rather than per-comparison, as our evaluation employs an MLLM-as-a-judge protocol, where a single language model provides preference judgments. For each combination (task $\times$ model pair), we query the judge $N$ times to obtain a stable estimate of its preference. Crucially, these $N$ responses are not independent samples from a population of raters—they are repeated stochastic draws from a single judge with fixed underlying preferences. The variation across repetitions reflects only sampling noise (due to temperature), not genuine diversity of opinion. Therefore, majority voting is the more sensible choice as it treats repeated queries as a variance-reduction mechanism, rather than as independent observations.

## H. Details of User Study

This section details the experimental protocols of our user study. We used Mabyduck[1] to conduct human evaluation and, in total, hired 150 unique crowd-sourced raters from Prolific. Figures 7 and 8 exemplify the user interface presented to the raters.

### H.1. Absolute Rating User Study Details

For our absolute score ratings, we performed three separate experiments, each focusing on just one semantic metric (*i.e.*, one of Realism, Functionality, and Instruction Following). This helps reduce the cognitive load on the raters and potentially allows for higher-quality results. For each category of rating, we recruited 30 crowd-sourced participants, and

Similarly to G.3, we query Gemini 3 Flash three times, each time providing the top-down and side-view renderings of two layouts generated by two methods on the same task and asking it to choose a preferred layout. The ordering of the methods (*i.e.*, which method's layout is placed on which side) is randomized at each step. Then, the three results are aggregated via majority voting.

This protocol yields a win/lose matrix between each pair of methods across all 27 tasks. A Bradley-Terry model is then fit on this task-level win/lose matrix via Sequential Least Squares Programming (SLSQP). Then, the estimated ratings $\hat{r}_i$ are converted to the traditional Elo scale via

$$\text{Elo}_i = 1500 + \frac{400}{\ln(10)} \cdot \hat{r}_i \qquad (48)$$

---

[1]www.mabyduck.com

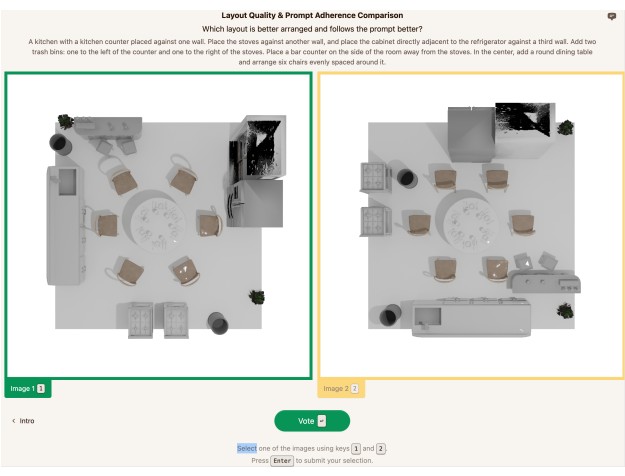

*Figure 8.* User study interface for pairwise rating.

*Table 9.* **Runtime comparison.** Average runtime (minutes) per scene for different methods under the same settings.

| Method | Runtime (min) |
| --- | --- |
| LayoutGPT | 3.6 |
| Holodeck | 3.2 |
| LayoutVLM | 5.0 |
| **R³L (Ours)** | 4.8 |

fully represent non-convex geometry. In addition, relations such as inside require explicit containment and clearance reasoning that goes beyond the current OBB formulation. Supporting such relations would likely require mesh-level or SDF-based representations, which we leave to future work.

**(3) Retrieval asset library.** Our framework relies on a predefined asset library, as retrieval provides editable assets with clean geometry and decomposable materials. However, scene diversity, fidelity, and object availability are bounded by the coverage of the asset pool. Future advances in 3D generation, especially for articulated and geometrically diverse objects, could help mitigate this limitation.

## K. Additional Experimental Results

### K.1. Runtime Comparison

We provide an additional runtime comparison in Table 9. Although LayoutGPT and Holodeck are 25% and 33% faster, they achieve substantially weaker performance: compared to LayoutGPT, R³L reduces %CR/%OR from 4.3/8.1 to 0.0/0.0; compared to Holodeck, it improves Real./Func./Instr. by about 100%/130%/200%. Compared to LayoutVLM, R³L is slightly faster while improving Real./Func./Instr. by about 60%/55%/30%. These results suggest a favorable trade-off between quality and efficiency.

### K.2. Comparison with Visual-Intermediate Methods

LLM-based methods (Yang et al., 2024b; Sun et al., 2025) and visual-intermediate methods (Ling et al., 2025; Wang et al., 2024) address the problem from different perspectives. Visual-intermediate approaches primarily emphasize visual fidelity, whereas our method focuses on spatial correctness through controllable and instruction-faithful relations.

To further examine this, we construct visual-intermediate baselines following (Wang et al., 2024) by first generating 2D layout images using two image generation backbones (GPT-Image-1.5 and Nano Banana) and then lifting them back to 3D using grounding and depth estimation models. To provide a fair upper-bound estimate, human annotators further correct misidentified or misplaced objects in the lifted results by referencing the 2D images.

As shown in Table 10 and Figure 11, R³L outperforms these

each participant was asked to rate the scene on a 5-point scale (corresponding to scores of 1–5). We then rescaled the results by uniformly mapping $[1, 5] \to [1, 10]$ to ensure consistency with Table 1.

### H.2. Pairwise User Study Details

For our pairwise user study in Table 5, 60 unique participants were recruited, each of whom was shown 27 pairs of layouts. Participants were asked to select their preferred layout on the interface shown in Figure 8. In total, this gives us 1,620 votes that uniformly cover all tasks × method pair combinations. To ensure consistency with G.4 and thus interpretability, we report the win counts at task-level and fit a Bradley-Terry model with identical settings.

## I. Failure Cases

We present representative failure cases in Figure 9 to better characterize the limitations of R³L. Common issues include a navigability gap in collision-free layouts, where clearance remains insufficient for human passage, and errors under complex rotational configurations, where dense layouts with diverse orientations make rotation reasoning less reliable.

## J. Limitations and Future Work

**(1) Planar pose formulation.** We restrict our study to 2D planar poses to isolate the core challenge of relative spatial reasoning. This formulation does not support richer 3D spatial relations such as on top of or multi-level layouts, which require reasoning over support surfaces at varying heights. Extending the framework beyond planar, ground-level layouts is an important direction for future work.

**(2) Bounding-box geometry.** Our optimization operates on Oriented Bounding Boxes (OBBs), which cannot faith-

*Table 10.* Comparison with visual-intermediate methods. Following ([Wang et al., 2024](#)), we generate 2D layout images using image generation models and lift them to 3D. R³L outperforms visual-intermediate baselines in both physical and semantic metrics.

| Method | %CR↓ | %OR↓ | Real.↑ | Func.↑ | Instr.↑ |
|---|---|---|---|---|---|
| Visual (GPT-Image-1.5) | 2.9 | 2.7 | 5.6 | 4.8 | 5.0 |
| Visual (Nano Banana) | 4.4 | 0.7 | 5.5 | 4.8 | 5.3 |
| **R³L (Ours)** | **0.0** | **0.0** | **7.9** | **7.5** | **9.1** |

*Table 11.* Generality across different backbones. Quantitative results averaged across all scene types. We report results for all four methods under two MLLM backbones. Lower is better for %CR/%OR, and higher is better for Real./Func./Instr.

| Backbone | Method | %CR↓ | %OR↓ | Real.↑ | Func.↑ | Instr.↑ |
|---|---|---|---|---|---|---|
| | LayoutGPT | 4.3 | 8.1 | 5.9 | 5.5 | 8.1 |
| GPT-5 | Holodeck | 1.3 | 0.8 | 4.0 | 3.3 | 3.0 |
| | LayoutVLM | 0.6 | 10.9 | 4.9 | 4.8 | 6.9 |
| | **R³L (Ours)** | **0.0** | **0.0** | **7.9** | **7.5** | **9.1** |
| | LayoutGPT | 4.2 | 7.6 | 6.2 | 6.0 | 8.4 |
| Gemini 3.1 Pro | Holodeck | 1.2 | 0.4 | 4.2 | 3.1 | 3.2 |
| | LayoutVLM | 0.8 | 10.0 | 5.1 | 5.2 | 7.3 |
| | **R³L (Ours)** | **0.0** | **0.0** | **8.0** | **7.7** | **9.2** |

*Table 12.* Anchor selection stability analysis. Each configuration is run 9 times per task. For each task, we compute the standard deviation of each metric across runs, and report RMS-averaged values over all tasks. Lower values indicate greater stability.

| Method | Average Std. Dev. (↓) | | | | |
|---|---|---|---|---|---|
| | %CR | %OR | Real. | Func. | Instr. |
| *w/o* Decomp. | 0.83 | 1.63 | 1.331 | 1.255 | 0.957 |
| **R³L (Ours)** | **0.31** | **0.90** | **0.848** | **0.908** | **0.945** |

baselines in both physical feasibility and semantic coherence. These results suggest that our LLM-based method is better suited for tasks requiring precise spatial control and faithful instruction following.

### K.3. Generality across Backbones

To evaluate generality across backbones, we provide additional results using Gemini 3.1 pro in Table 11 and Figure 15. R³L consistently outperforms the corresponding baselines, indicating that R³L is not tied to GPT-5 alone.

### K.4. Anchor Selection

We do not impose a hand-crafted anchor selection criterion. Instead, the MLLM determines the anchor jointly with unit construction, with the only constraint that the anchor connects the unit to the rest of the scene.

To examine the sensitivity of this design, we conduct repeated experiments under the same setting in Table 12. The full model exhibits consistently lower run-to-run variance than the variant without decomposition. These results suggest that decomposition with anchor assignment improves the stability of the overall framework.

### K.5. CLIP/BLIP/VQA Scores

Following ([Ling et al., 2025](#)), we additionally report CLIP, BLIP and VQA scores in Table 13. R³L achieves the highest average scores across these metrics, although the ranking of methods remains unstable across different scene types.

We further evaluate the alignment of these metrics with human judgments in Table 14. The correlations are notably limited for CLIP/BLIP/VQA, performing substantially weaker than our LLM-based evaluation. A likely explanation is that these traditional metrics primarily capture global image-text semantic compatibility rather than the precise instantiation of specific spatial relations. Consequently, we include CLIP/BLIP/VQA scores as supplementary metrics, while retaining LLM-based evaluation as the primary metric for fine-grained spatial correctness. Together, they provide a more comprehensive assessment of the generated layouts.

### K.6. MLLM Reasoning Trace Example

Figure 10 shows an example of an MLLM reasoning trace, illustrating how repeated reference-frame shifts in spatial reasoning lead to semantic drift.

### K.7. Placement Error Rate by Hop Count

We report the percentage of misplaced objects, grouped by their hop count from the root of the inferred relation graph. For each rendered layout, two blinded human annotators first identify misplaced objects, and the marked objects are then assigned to bins based on their hop count in the relation graph. As shown in Table 15, when both modules are removed, the error rate increases substantially with hop count. Our full model maintains low error rate across all hop counts, reducing both the overall error rate and its growth.

*Table 13.* Quantitative results of CLIP/BLIP/VQA scores for 3D layout generation across nine scene types. CLIP score is computed as the cosine similarity between image and text embeddings. BLIP score is obtained from the BLIP-2 ITM head. VQA score is computed using the CLIP-FlanT5 backbone. Following (Ling et al., 2025), we average results across two different views to reduce view-dependent bias.

| Method | Bathroom | | | Bedroom | | | Bookstore | | | Game Room | | | Gym | | |
|---|---|---|---|---|---|---|---|---|---|---|---|---|---|---|---|
| | CLIP↑ | BLIP↑ | VQA↑ | CLIP↑ | BLIP↑ | VQA↑ | CLIP↑ | BLIP↑ | VQA↑ | CLIP↑ | BLIP↑ | VQA↑ | CLIP↑ | BLIP↑ | VQA↑ |
| LayoutGPT | 22.86 | 22.72 | 0.7973 | 27.74 | 48.65 | 0.7722 | 21.17 | 37.70 | 0.8537 | 24.32 | 53.08 | 0.8687 | 27.16 | 64.65 | 0.8473 |
| Holodeck | **24.75** | 12.57 | 0.8205 | **28.81** | 51.33 | **0.8325** | **21.63** | 40.88 | **0.9163** | 24.07 | 39.62 | 0.8434 | **30.97** | **77.95** | 0.8779 |
| LayoutVLM | 21.47 | **27.79** | 0.7775 | 28.58 | **54.66** | 0.7520 | 21.09 | 39.19 | 0.8912 | 23.25 | 50.79 | 0.8730 | 30.00 | 75.13 | 0.8223 |
| **R³L (Ours)** | 24.26 | 27.19 | **0.8390** | 28.33 | 45.37 | 0.7457 | 21.55 | **52.15** | 0.9043 | **25.32** | **58.29** | **0.8864** | 30.50 | 65.68 | **0.8908** |

| Method | Kitchen | | | Living Room | | | Music Room | | | Restaurant | | | Average | | |
|---|---|---|---|---|---|---|---|---|---|---|---|---|---|---|---|
| | CLIP↑ | BLIP↑ | VQA↑ | CLIP↑ | BLIP↑ | VQA↑ | CLIP↑ | BLIP↑ | VQA↑ | CLIP↑ | BLIP↑ | VQA↑ | CLIP↑ | BLIP↑ | VQA↑ |
| LayoutGPT | 24.70 | 38.26 | 0.8935 | **26.56** | 66.84 | **0.8960** | 30.28 | 84.77 | 0.8974 | **32.42** | 81.88 | 0.7351 | 26.36 | 55.39 | 0.8401 |
| Holodeck | 24.33 | 39.48 | 0.8959 | 23.94 | 64.88 | 0.8867 | 26.14 | 77.37 | 0.9143 | 29.33 | 82.85 | 0.7810 | 26.00 | 54.11 | 0.8632 |
| LayoutVLM | 22.60 | 49.82 | 0.8594 | 24.58 | **66.86** | 0.8936 | 27.83 | 76.94 | 0.9027 | 30.90 | **86.93** | 0.8051 | 25.59 | 58.68 | 0.8419 |
| **R³L (Ours)** | **25.07** | **53.10** | 0.8855 | 25.95 | 61.59 | 0.8908 | 28.73 | **89.21** | **0.9210** | 30.32 | 86.33 | **0.8122** | **26.67** | **59.88** | **0.8640** |

*Table 14.* Alignment between human judgments and CLIP/BLIP/VQA scores. We compute Kendall's $\tau$-b between the rankings induced by human judgments and CLIP/BLIP/VQA scores. The substantially weaker correlations of CLIP/BLIP/VQA indicate that these metrics serve only as supplementary signals for fine-grained spatial correctness in our setting.

| Method | Real. ↑ | Func. ↑ | Instr. ↑ |
|---|---|---|---|
| User-User | **0.56** | **0.55** | 0.53 |
| User-LLM | 0.52 | 0.50 | **0.56** |
| User-CLIP | 0.08 | 0.11 | 0.09 |
| User-BLIP | 0.03 | 0.07 | 0.04 |
| User-VQA | 0.08 | 0.07 | -0.01 |

*Table 15.* Placement error rate by hop count (% ↓). **Adding Decomp. noticeably flattens the growth** by reducing reference-frame shifts. **Adding Imag. lowers the overall level** by improving metric consistency.

| Method | Hop 1 | Hop 2 | Hop 3 | Hop 4 | Hop 5+ |
|---|---|---|---|---|---|
| *w/o* Decomp. & Imag. | 14.8 | 22.3 | 22.7 | 30.5 | 47.6 |
| *w/o* Decomp. | 5.7 | 12.3 | 15.0 | 24.5 | 31.2 |
| *w/o* Imag. | 11.9 | 18.5 | 15.0 | 20.8 | 22.4 |
| **R³L (Ours)** | **3.5** | **9.6** | **11.2** | **6.9** | **12.3** |

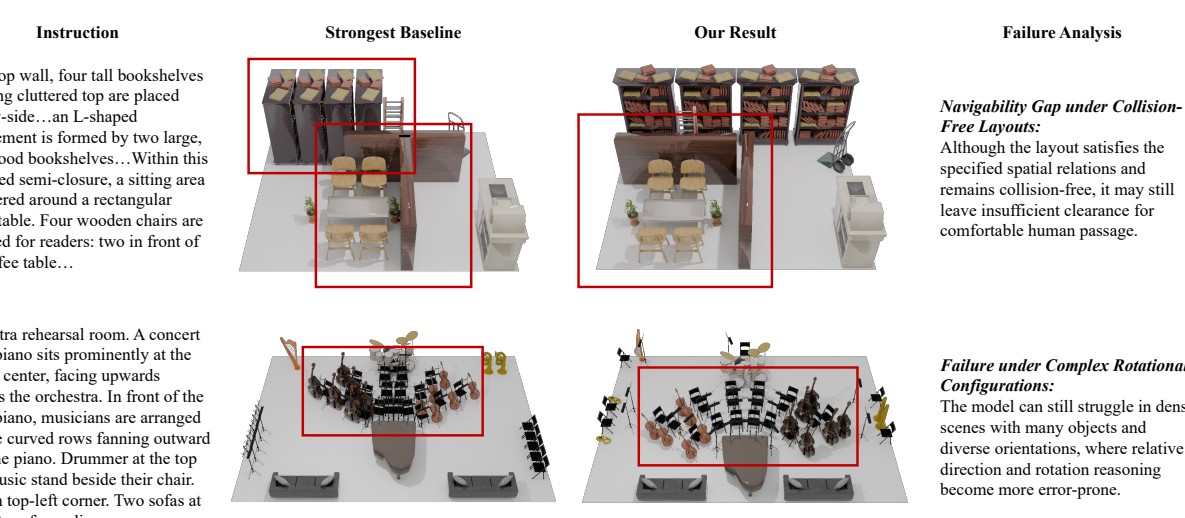

| Instruction | Strongest Baseline | Our Result | Failure Analysis |
|---|---|---|---|
| …the top wall, four tall bookshelves featuring cluttered top are placed side-by-side…an L-shaped arrangement is formed by two large, dark wood bookshelves…Within this L-shaped semi-closure, a sitting area is centered around a rectangular coffee table. Four wooden chairs are arranged for readers: two in front of the coffee table… | | | ***Navigability Gap under Collision-Free Layouts:*** Although the layout satisfies the specified spatial relations and remains collision-free, it may still leave insufficient clearance for comfortable human passage. |
| Orchestra rehearsal room. A concert grand piano sits prominently at the bottom center, facing upwards towards the orchestra. In front of the grand piano, musicians are arranged in three curved rows fanning outward from the piano. Drummer at the top with music stand beside their chair. Harp in top-left corner. Two sofas at the bottom for audience. | | | ***Failure under Complex Rotational Configurations:*** The model can still struggle in dense scenes with many objects and diverse orientations, where relative direction and rotation reasoning become more error-prone. |

*Figure 9.* **Representative failure cases.** Navigability gaps in collision-free layouts and errors under complex rotational configurations.

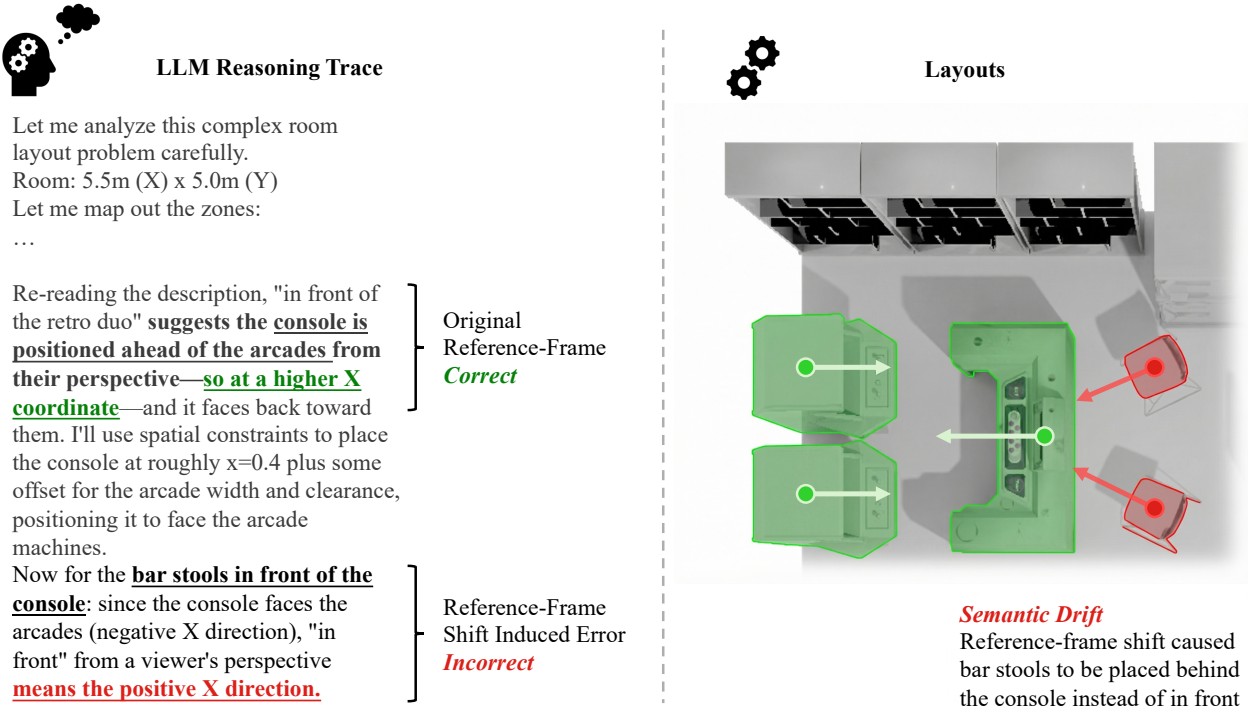

**LLM Reasoning Trace**

Let me analyze this complex room layout problem carefully.
Room: 5.5m (X) x 5.0m (Y)
Let me map out the zones:
…

Re-reading the description, "in front of the retro duo" **suggests the console is positioned ahead of the arcades from their perspective—so at a higher X coordinate**—and it faces back toward them. I'll use spatial constraints to place the console at roughly x=0.4 plus some offset for the arcade width and clearance, positioning it to face the arcade machines.

Original Reference-Frame
*Correct*

Now for the **bar stools in front of the console**: since the console faces the arcades (negative X direction), "in front" from a viewer's perspective **means the positive X direction.**

Reference-Frame Shift Induced Error
*Incorrect*

**Layouts**

*Semantic Drift*
Reference-frame shift caused bar stools to be placed behind the console instead of in front

*Figure 10.* MLLM reasoning trace example. The model initially places the console correctly in front of the arcade machines, but after a reference-frame shift, it misinterprets "in front of the console" and places the bar stools behind the console.

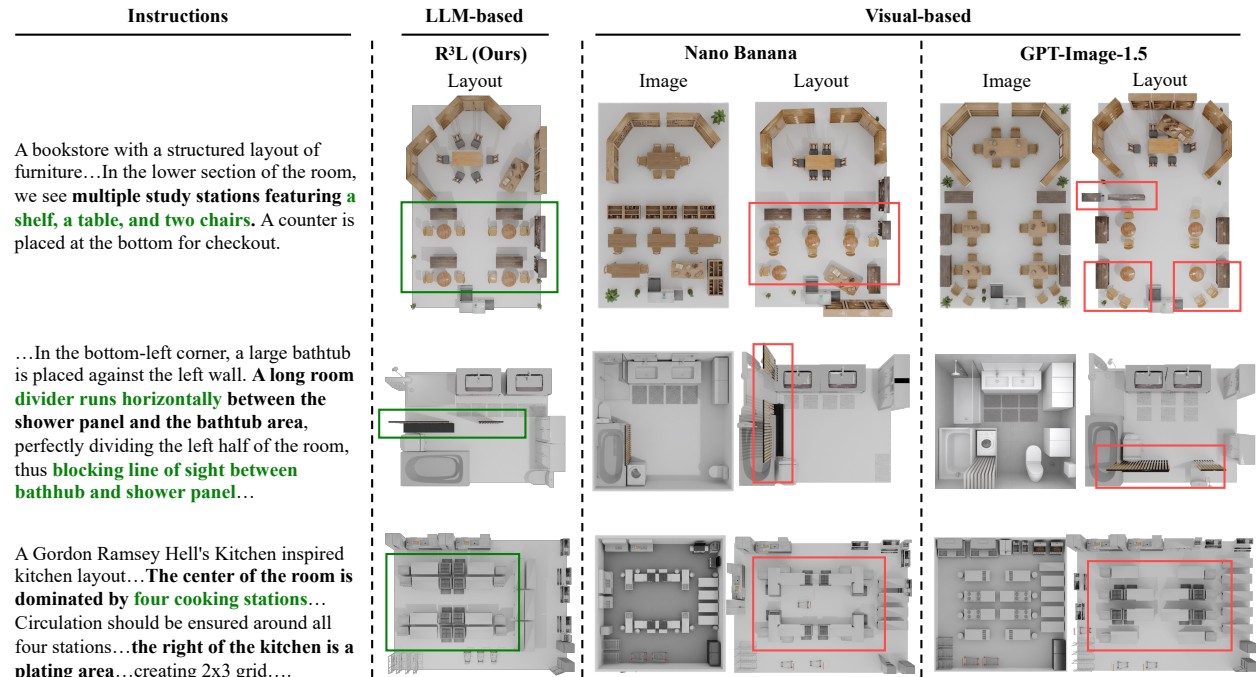

*Figure 11.* Qualitative comparison with visual-intermediate methods. Compared to visual-intermediate baselines, R³L better preserves fine-grained instruction following and explicit spatial correctness.

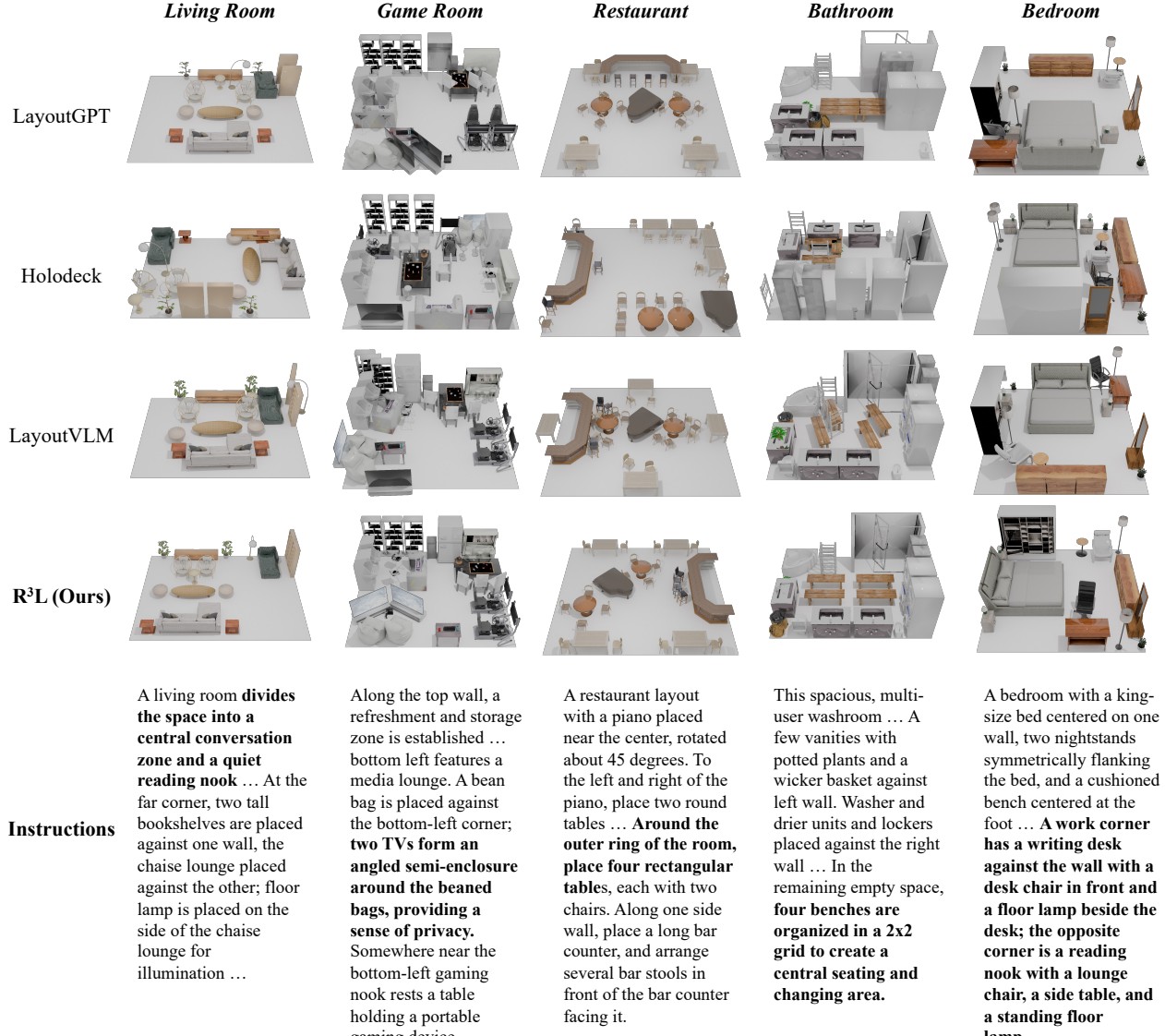

|  | Living Room | Game Room | Restaurant | Bathroom | Bedroom |
|---|---|---|---|---|---|

**LayoutGPT**

**Holodeck**

**LayoutVLM**

**R³L (Ours)**

**Instructions**

A living room **divides the space into a central conversation zone and a quiet reading nook** … At the far corner, two tall bookshelves are placed against one wall, the chaise lounge placed against the other; floor lamp is placed on the side of the chaise lounge for illumination …

Along the top wall, a refreshment and storage zone is established … bottom left features a media lounge. A bean bag is placed against the bottom-left corner; **two TVs form an angled semi-enclosure around the beaned bags, providing a sense of privacy.** Somewhere near the bottom-left gaming nook rests a table holding a portable gaming device.

A restaurant layout with a piano placed near the center, rotated about 45 degrees. To the left and right of the piano, place two round tables … **Around the outer ring of the room, place four rectangular table**s, each with two chairs. Along one side wall, place a long bar counter, and arrange several bar stools in front of the bar counter facing it.

This spacious, multi-user washroom … A few vanities with potted plants and a wicker basket against left wall. Washer and drier units and lockers placed against the right wall … In the remaining empty space, **four benches are organized in a 2x2 grid to create a central seating and changing area.**

A bedroom with a king-size bed centered on one wall, two nightstands symmetrically flanking the bed, and a cushioned bench centered at the foot … **A work corner has a writing desk against the wall with a desk chair in front and a floor lamp beside the desk; the opposite corner is a reading nook with a lounge chair, a side table, and a standing floor lamp** …

*Figure 12.* Additional qualitative comparisons between R³L and existing baselines under the same instructions. R³L consistently produces physically feasible and semantically consistent layouts across tasks, while effectively following instructions.

**DETAILED**

**CONCISE**

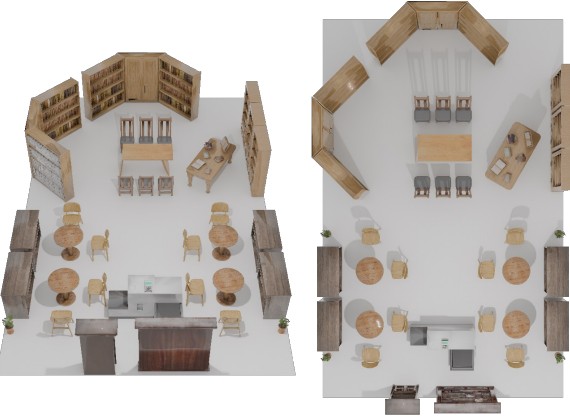 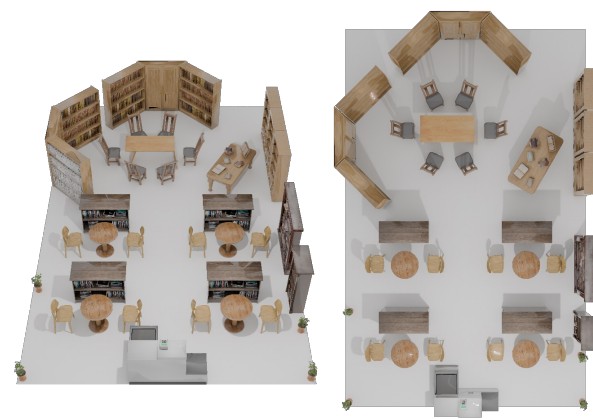

A bookstore with a structured layout of furniture. In the upper-central area of the room starts with two large carved-front V-shaped bookshelves, one against top wall, one against left wall, their corners precisely touching … In the lower section of the room, two tight and intimate reading stations occupy opposite walls. Within **each station, a pair of shelves placed side-by-side flanked by two potted plants; each shelf has a round table in front facing same direction. The station's left-side round table has two curved-back wooden chairs on its front and left, the right-side table also has two chairs but on its front and right. The two study stations are vertically aligned but against opposite walls, creating symmetry.** Finally, two wooden bookshelves of varying sizes centered against bottom wall with a checkout counter closely in front of it.

A bookstore with a structured layout of furniture. In the upper-central area of the room starts with two large carved-front V-shaped bookshelves, one against top wall, one against left wall, their corners precisely touching. Within these V-shaped bookshelves rests a rectangular wooden table with high backrest wooden chairs around it. **In the lower section of the room, there are multiple study stations featuring a shelf, a table, and two chairs. A counter is placed at the bottom for checkout.**

*Figure 13.* Qualitative results demonstrating robustness to varying levels of prompt verbosity under the same asset set and room dimensions. **Left:** R³L accurately reproduces the complex spatial arrangement of *study stations* from a detailed prompt. **Right:** R³L produces equally compelling layouts from a high-level description (*i.e.*, "multiple study stations featuring a shelf, a table, and two chairs").

| Inputs | Invariant Spatial Decomposition | Consistent Spatial Imagination | Supportive Spatial Optimization |
|---|---|---|---|
| A sofa is **in front of** and **facing** a TV stand. Coffee table is **in front of** sofa. Lounge chairs are **near** coffee table… | **Unit 1:** Sofa
**Unit 2:** TV Stand
**Unit 3:** Table (anchor) + Chair 1/2

**Intra-Unit Relations:**
Chair 1/2 *face to* (0.1m) Table
**Inter-Unit Relations:**
Unit 1 *in front of* (4m) Unit 2
Unit 3 *in front of* (1m) Unit 1 | **Local Map** (Unit 3)
Table (0, 0)    Table (0, 0)
Chair 1 (0.1, 0) → Chair 1 (0.5, 0)
Chair 2 (0, 0.1)   Chair 2 (0, 0.5)
**Global Map**
Unit1 (0, 0)
Unit 2 (0, 4.0)   Chair 1/2 *face to*
Unit 3 (0, 1.0)   (0.5m) Table
*Red: Collision*   *Green: Consistent* | **Intra-Unit Relations:**
Chair 1/2 *face to* (0.5m) Table
**Inter-Unit Relations:**
Unit 1 *in front of* (4m) Unit 2


Layouts |

*Figure 14.* A run-through example of R³L that illustrates how R³L performs decomposition, imagination, and optimization.

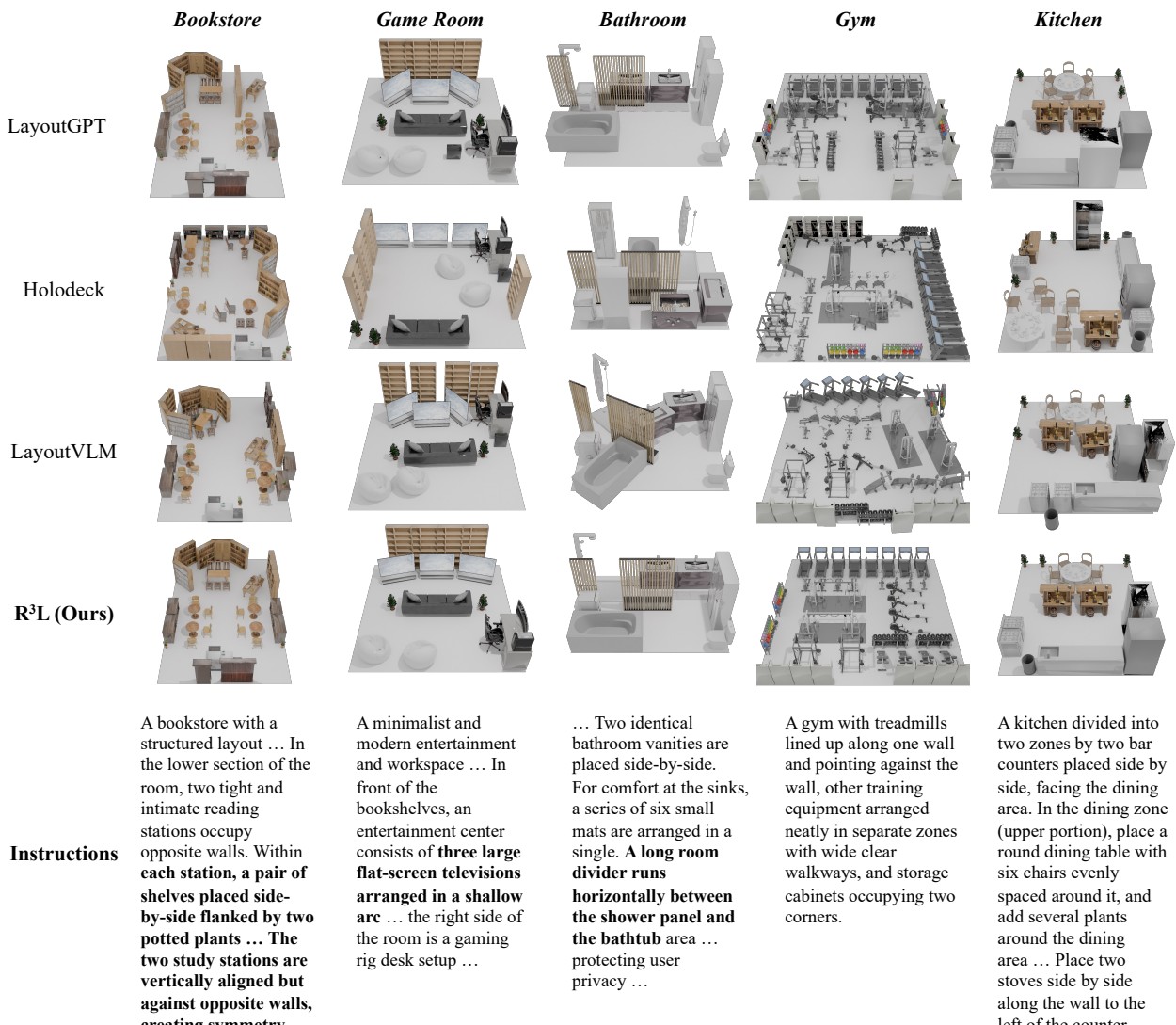

*Figure 15.* Qualitative results using Gemini 3.1 pro as the backbone. R³L consistently improves over the corresponding Gemini-based baseline, showing that the proposed framework is not tied to GPT-5 and generalizes to a different MLLM backbone.

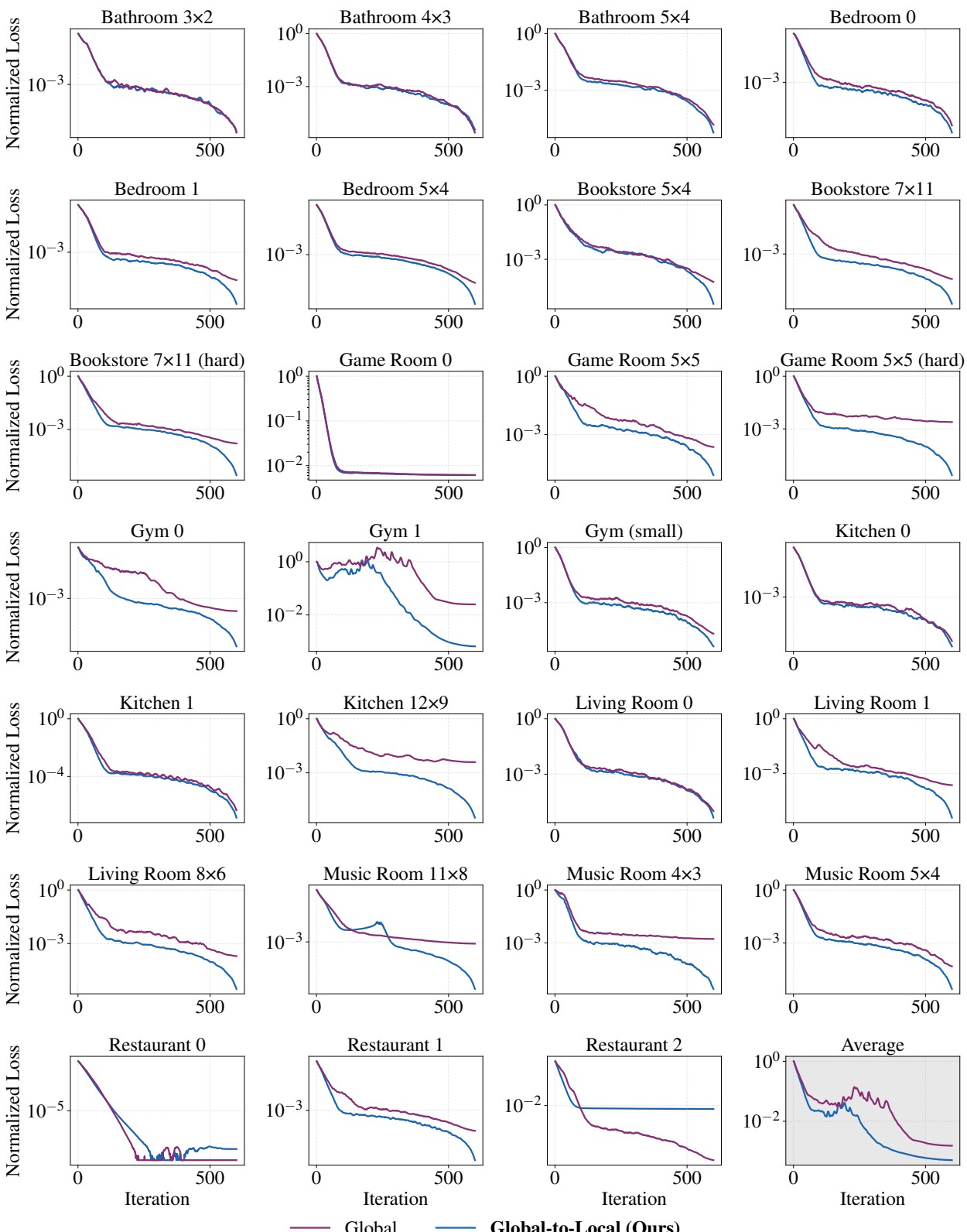

*Figure 16.* Comparison of convergence curves for global and global-to-local optimization under identical optimizer settings. Each plot is obtained by averaging the normalized loss of three independent runs using different seeds and smoothed with EMA ($\alpha = 0.85$). Overall, global-to-local reparameterization achieves faster convergence and lower final loss in scenes with a large number of objects.

