# OpenReview forum: "R$^3$L: Reasoning 3D Layouts from Relative Spatial Relations"
_ICML.cc/2026/Conference — ICML 2026 regular_

### Official Review · Reviewer_QeVV · 2026-02-21

**Soundness:** 2
**Presentation:** 3
**Significance:** 2
**Originality:** 3
**Overall Recommendation:** 4
**Confidence:** 4

**Summary:**

This paper studies 3D layout generation from language-specified relative spatial relations (e.g., “chair in front of table”) and points out that existing MLLM-based methods often chain many pairwise relations, which causes reference-frame shifts and leads to semantic drift and physically invalid layouts. The authors propose R³L, a framework that decomposes scenes into frame-invariant units and breaks long relation chains into shorter sub-chains to reduce accumulated errors; uses a consistent spatial imagination loop where an MLLM imagines, checks, and revises object placements for self-consistency, and applies supportive spatial optimization with a global-to-local pose reparameterization to stabilize layout optimization.

**Compliance With Llm Reviewing Policy:**

Affirmed.

**Final Justification:**

The study addresses an important problem in 3D scene generation. The authors propose R³L, a framework that decomposes scenes into frame-invariant units and breaks long relation chains into shorter sub-chains, thereby reducing the accumulation of errors in language-specified relative spatial relations. I initially had concerns about the motivation and experimental details. However, the authors carefully addressed most of these concerns and provided sufficiently thorough analysis of the experiments. I encourage the authors to incorporate these additional experiments and the discussion of limitations into the final version. Overall, I would increase my rating to Weak Accept.

**Key Questions For Authors:**

Please see what I mentioned in the weaknesses section.

**Limitations:**

Yes

**Strengths And Weaknesses:**

Strengths:
1. The paper gives the analysis of how multi-hop relation chains and repeated reference-frame shifts cause semantic and metric drift in 3D layout reasoning;
2. The proposed combination of (i) invariant spatial decomposition, (ii) consistent spatial imagination (imagine-and-revise with explicit cognitive maps), and (iii) supportive spatial optimization with global-to-local re-parameterization is conceptually clean, modular, and reasonable to instruction-driven 3D layout tasks.

Weaknesses:
1. I did not get the importance of the multi-hop relation chains for LLM-based planning. How does it performanced compared to these visual intermidiate method?
1. Unclear importance of multi-hop chains. The paper argues that multi-hop relation chains are the core bottleneck for LLM-based method, but does not compare against methods that also LLM-based while use visual as intermediate supervision to address spatial relations (e.g., these discussed in the related work section), so the practical significance of this focus is not fully demonstrated. Besides, how does the method handle scenes with multiple identical objects (e.g., several desks)? LLMs are known to hallucinate or be unreliable with numerical reasoning and counting, so disambiguating which specific instance each instruction refers to may be challenging.
2. Limited, indirect semantic evaluation & practical characterization.
Semantic quality is mainly measured via LLM- and human-rated holistic scores (Real/Func/Instr). The paper does not report fine-grained relational accuracy such as CLIP based text following metrics, making it harder to assess performance and robustness.
3. There is no disussion of the limitation.
The method operates on 2D planar poses with axis-aligned bounding boxes, which simplifies optimization but cannot faithfully represent non-convex object geometries or richer 3D relations such as “inside”, “on top of a shelf”, or multi-level layouts. This limits applicability to more complex 3D scenes and spatial relations.
5. No runtime comparison with baselines or failure analysis. Beyond a convergence plot for their optimizer, the paper does not report runtime or failure cases compared to baselines, so the practical efficiency and robustness remain unclear.

---

> ### Author Rebuttal · Authors · 2026-03-31
>
> We appreciate the reviewer's thoughtful comments and address all concerns below. An anonymous PDF with figures and tables is available at: https://anonymous.4open.science/r/ICML4647-BF6F/ICML_4647_rebuttal.pdf
>
> # Q1: Importance of Multi-hop Chains
>
> A1: Thanks for your question. In LLM-based methods, when inferring relative spatial relations, multi-hop reasoning is unavoidable whenever two objects are connected through intermediate objects. In these multi-hop chains, previously inferred relations must be repeatedly re-expressed across different frames, making error accumulation a challenge. To support this, the attached ***Table 1*** and ***Figure 1*** show that the placement error rate generally increases with the number of hops (i.e., reference-frame shifts), from **14.8%** at **Hop 1** to **47.6%** at **Hop 5+** when our modules are removed. In contrast, our full model consistently achieves much lower errar rates (**3.5%-12.3%**) across all hop distances. We will clarify this in the revision.
>
> # Q2: Visual-Intermediate Methods
>
> A2: We would like to clarify that LLM-based relational reasoning and visual-intermediate methods address the problem from different angles. Visual-intermediate methods mainly focus on visual fidelity, whereas our method focuses on spatial correctness through controllable and instruction-faithful relations.
>
> To further examine this, we construct visual-intermediate baselines following Architect. As shown in the attached ***Table 4*** and ***Figure 5***, R$^{3}$L outperforms visual-intermediate baselines in both physical feasibility and semantic performance, reducing **%CR/%OR from 2.9/2.7 and 4.4/0.7 to 0.0/0.0** and improving **Real./Func./Instr.** from **5.6/4.8/5.0** and **5.5/4.8/5.3** to **7.9/7.5/9.1**. These results suggest that our LLM-based method is better suited to our task of generating reliable, controllable, and instruction-faithful layouts.
>
> # Q3: Identical Objects
>
> A3: For scenes with multiple identical objects, we explicitly disambiguate them at the instance level. In our Python DSL, each object instance is assigned a unique instance_id, and all spatial relations are grounded to these instance-level references. Our DSL parser further checks for missing references and prompts a revision if any are detected. In practice, such cases are rare. An illustrative example is shown in the attached ***Figure 6***.
>
> # Q4: CLIP-based Evaluation
>
> A4: We provide additional CLIP-based scores in the attached ***Table 5***. While our method achieves the highest average CLIP score, the ranking of methods is not stable across scene types. We further evaluate the alignment between CLIP scores and human judgments (attached ***Table 6***), which shows only limited correlation (**0.08/0.11/0.09**) and is clearly weaker than that of LLM-based evalution (**0.52/0.50/0.56**). This is expected, as CLIP mainly measures coarse global text-scene compatibility, whereas our task requires fine-grained evaluation of spatial relation correctness. Therefore, we regard CLIP score as a supplementary metric rather than a primary one. We will clarify this in the revision.
>
> # Q5: Planar Pose Limitation
>
> A5: We agree that the current planar formulation is a limitation. As stated in Section 3.1, we restrict our study to 2D planar poses to isolate the core challenge of relative spatial reasoning. We also clarify that our optimization uses OBBs rather than AABBs.
>
> For non-convex objects, the current box-based geometry can be extended to richer representations such as SDFs without changing the core optimization idea. For richer 3D relations, a natural extension is to perform hierarchical decomposition over support surfaces and re-apply our framework on each surface, as in HSM. In this sense, the key idea of our optimization, reducing local-global gradient coupling, remains unchanged. We will add this discussion of limitation in the revision.
>
> # Q6: Runtime and Failure Cases
>
> A6: We provide an additional runtime comparison in the attached ***Table 7***. Although LayoutGPT and Holodeck are 25% and 33% faster, they achieve substantially weaker performance: compared to LayoutGPT, R$^{3}$L reduces **%CR/%OR from 4.3/8.1 to 0.0/0.0**; compared to Holodeck, it improves **Real./Func./Instr.** by about **100%/130%/200%**. Compared to LayoutVLM, R$^{3}$L is slightly faster while improving **Real./Func./Instr.** by about **60%/55%/30%**. This shows a stronger quality-efficiency trade-off.
>
> We also provide representative failure cases (attached ***Figure 7***) to better characterize the limitations of R$^{3}$L. Typical remaining issues include a navigability gap in some collision-free layouts, where clearance is insufficient for human passage, and errors under complex rotational configurations, where dense layouts with diverse orientations make rotation reasoning less reliable. We will discuss these failure cases in the revision.

---

> > ### Author Rebuttal · Reviewer_QeVV · 2026-04-05
> >
> > I appreciate author's rebuttal and the authors have addressed my concerns on baseline comparsion with vision-intermediate method, importance of multi-hop chains, and running time. I still have following questions:
> > 1. Quantitative evaluation. As the CLIP score might not be suitable to quantify fine-grained spatial relations, there are other metrics the author might want to use such as BLIP score and VQA.
> > 2. Limitation. The authors reasonably clarify that the planar formulation is an intentional scope choice rather than an oversight, and the OBB clarification is helpful. Their proposed extensions to richer geometry and hierarchical support surfaces are plausible. However, these are future directions rather than evidence that the current framework can already handle non-convex geometry or richer 3D spatial relations. In particular, relations such as inside require containment and clearance reasoning beyond the proposed support-surface extension.  More broadly, it would also be helpful for the paper to discuss other practical limitations, such as depends on a predefined 3D asset library (e.g., Objaverse) for object availability and object quality. As a result, scene diversity and fidelity may be bounded by the coverage of the asset pool. Thus, the limitation remains substantive. It would strengthen the paper if the authors explicitly discussed these boundaries of applicability in the limitations section.

---

> > > ### Author Response · Authors · 2026-04-07
> > >
> > > Dear Reviewer QeVV,
> > >
> > > Thank you for the follow-up and for acknowledging that our rebuttal has addressed your concerns regarding the comparison with vision-intermediate methods, the importance of multi-hop relation chains, and runtime. We address your additional concerns below.
> > >
> > > An anonymous PDF with tables is available at: https://anonymous.4open.science/r/ICML_4647-E4D3/ICML_4647_rebuttal_2.pdf
> > >
> > > # Q1: BLIP and VQA Scores
> > >
> > > A1: Thank you for this suggestion. Following *Scenethesis*, we additionally report **CLIP/BLIP/VQA** scores in the attached ***Table 10***. Our method achieves the highest average CLIP/BLIP/VQA scores (**26.67/59.88/0.8640**), although the ranking of methods is not stable across scene types. We further evaluate how these metrics align with human judgments (attached ***Table 11***). Their correlations with human judgments are limited for CLIP (**0.08/0.11/0.09**), BLIP (**0.03/0.07/0.04**), and VQA (**0.08/0.07/-0.01**), and are substantially weaker than those of our LLM-based evaluation (**0.52/0.50/0.56**). A likely explanation is that these metrics primarily capture global image-text semantic compatibility, rather than whether specific spatial relations are correctly instantiated. Based on these results, we will include CLIP/BLIP/VQA scores as supplementary metrics in the revision, while retaining LLM-based evaluation as the primary metric for fine-grained spatial correctness. Together, they provide a more comprehensive evaluation.
> > >
> > > # Q2: Limitation Discussion
> > >
> > > A2: Thank you for this important suggestion. We agree that the extensions discussed in our previous rebuttal are future directions rather than capabilities demonstrated by the current framework. We agree that these boundaries of applicability should be stated more explicitly. In the revision, we will add a dedicated **Limitations** section to clarify that:
> > >
> > > **(1) Planar pose formulation.** We restrict our study to *2D planar poses* to isolate the core challenge of relative spatial reasoning. This formulation does not support richer 3D spatial relations such as *on top of a shelf* or *multi-level layouts*, which require reasoning over support surfaces at varying heights. Extending the framework beyond planar ground-level layouts is an important direction for future work.
> > >
> > > **(2) Bounding-box geometry.** Our optimization operates on *Oriented Bounding Boxes (OBBs)*, which cannot faithfully represent non-convex geometry. Moreover, relations such as *inside* require explicit containment and clearance reasoning that goes beyond the current OBB formulation. Supporting such relations would likely require mesh-level or SDF-based representations, which we leave to future work.
> > >
> > > **(3) Retrieval asset library.** Our framework uses a *predefined asset library* because retrieval provides editable assets with clean geometry and decomposable materials. However, scene diversity, fidelity, and object availability are bounded by the coverage of the asset pool. Future advances in 3D generation, especially for articulated and geometrically diverse objects, could help mitigate this limitation.
> > >
> > > We believe these clarifications will help readers better understand the scope and current boundaries of our framework. Thank you again for your constructive feedback throughout the review process.
> > >
> > > Best, Authors

---

### Official Review · Reviewer_Xqka · 2026-03-12

**Soundness:** 3
**Presentation:** 3
**Significance:** 2
**Originality:** 2
**Overall Recommendation:** 4
**Confidence:** 3

**Summary:**

This paper studies 3D layout generation from relative spatial relations and identifies error accumulation caused by repeated reference-frame switching in multi-hop spatial reasoning. To address this, the authors propose R3L, which combines invariant spatial decomposition, a self-consistent imagine-and-revise inference loop, and a global-to-local spatial optimization strategy. Experiments across diverse scene types show improved semantic consistency and physical feasibility compared with recent layout-generation baselines.

**Compliance With Llm Reviewing Policy:**

Affirmed.

**Final Justification:**

I will maintain my current positive score. No further comments at this time.

**Key Questions For Authors:**

1. Is the main bottleneck long relation-chain reasoning itself, or the repeated frame switching it induces?
2. How much of the improvement comes from invariant spatial decomposition versus the imagine-and-revise reasoning step?

**Limitations:**

The paper could more clearly discuss the generality of the approach across different backbones and scene distributions.

**Strengths And Weaknesses:**

- Identifies a concrete failure mode in spatial reasoning: error accumulation due to repeated frame switching in long relation chains.
- The method is conceptually coherent, with reasoning and optimization components both targeting the same drift issue.
- Experimental results are solid, showing improved semantic quality while maintaining or improving physical validity.

---

> ### Author Rebuttal · Authors · 2026-03-31
>
> We appreciate the reviewer's thoughtful comments and address all concerns below. An anonymous PDF with figures and tables is available at: https://anonymous.4open.science/r/ICML4647-BF6F/ICML_4647_rebuttal.pdf
>
> # Q1: Main Bottleneck
>
> A1: Thanks for your question. We respectfully clarify that the main bottleneck is the accumulated errors introduced by repeated reference-frame transformations during long multi-hop reasoning. Since relative spatial relations are defined in object-centric frames, each hop typically requires previously inferred relations to be re-expressed under a new local frame, allowing small inconsistencies to propagate into semantic and metric drift. The attached ***Table 1*** and ***Figure 1*** support this: the placement error rate increases with the number of hops (i.e., reference-frame shifts), from **14.8%** at **Hop 1** to **47.6%** at **Hop 5+** when our modules are removed, whereas our full model keeps the error rate much lower (**3.5%–12.3%**). We will clarify this in the revision.
>
> # Q2: Improvement from Different Modules
>
> A2: This is already supported by our ablation study (Table 3), which shows that invariant spatial decomposition and imagine-and-revise are both important and complementary. Relative to the full model, removing imagination increases **%CR** from **1.0 to 3.1** and **%OR** from **1.6 to 6.3**, while reducing **Real./Func./Instr.** from **8.0/7.3/9.1** to **6.2/6.0/7.9**. Removing decomposition increases **%CR** from **1.0 to 1.7** and **%OR** from **1.6 to 3.8**, while reducing **Real./Func./Instr.** to **6.9/6.7/8.5**. When both are removed, performance further drops to **3.1/7.9/5.7/5.4/7.8** on **%CR/%OR/Real./Func./Instr.**. These results suggest that both modules improve semantic quality and physical feasibility by producing more reliable relations.
>
> # Q3: Different Backbones and Scene Distributions
>
> A3: Thanks for this important comment. To evaluate generality across backbones, we provide additional results with Gemini 3.1 pro (attached ***Table 9*** and ***Figure 9***). R$^{3}$L shows consistent improvements over the corresponding baselines, indicating that the method is not tied to GPT-5 alone.
>
> For scene distributions, Table 1 already shows consistent gains across all nine scene types. The gains are especially clear in scenes such as **bookstore, living room, restaurant, and gym**, where layouts involve more coupled object relations. In contrast, in scenes such as **bedroom, game room, and music room**, the relative margin is smaller on some semantic metrics, suggesting that these scenes involve less severe multi-hop coupling. This pattern is consistent with our motivation: R$^{3}$L is most beneficial when multi-hop spatial reasoning and cross-object consistency are more challenging. We will add this cross-backbone and per-scene discussion in the revision.

---

### Official Review · Reviewer_VwH2 · 2026-03-12

**Soundness:** 2
**Presentation:** 2
**Significance:** 2
**Originality:** 2
**Overall Recommendation:** 3
**Confidence:** 5

**Summary:**

This paper studies instruction-driven 3D layout generation through the lens of relative spatial reasoning. The authors argue that existing MLLM-based pipelines often infer unreliable relative relations, especially in multi-hop reasoning chains where repeated reference-frame shifts can lead to semantic drift and metric drift. To address this, the paper proposes R3L, a two-stage framework with three main components: invariant spatial decomposition, which partitions a scene into frame-invariant units to shorten relation chains; consistent spatial imagination, which uses a local-to-global imagine-and-revise process to improve self-consistency and reduce geometric conflicts during relation inference; and supportive spatial optimization, which performs global-to-local pose re-parameterization to stabilize downstream optimization.

**Compliance With Llm Reviewing Policy:**

Affirmed.

**Final Justification:**

My questions were addressed to some extent

**Key Questions For Authors:**

See above in weakness.

**Limitations:**

yes

**Strengths And Weaknesses:**

Strengths:
1. The paper presents a clear and interesting perspective on 3D layout generation by focusing on the reliability of relative spatial reasoning rather than treating physical feasibility purely as a downstream solving issue. The decomposition of errors into semantic drift and metric drift gives the method a coherent high-level motivation.
2. The proposed method is technically structured and modular. The combination of invariant spatial decomposition, imagine-and-revise relation inference, and supportive spatial optimization is well aligned with the claimed problem setting.

Weakness:
1. The paper states that MLLMs often struggle to maintain consistent intermediate states during reasoning, but this claim is not sufficiently explained or empirically supported. While the paper attributes downstream failures to repeated reference-frame shifts, it remains unclear why MLLMs specifically fail to preserve consistent intermediate states in this setting, beyond the high-level intuition of multi-hop reasoning difficulty.
2. A key implementation detail remains unclear: how is the anchor selected for each unit? Since the anchor defines the unit-local frame, the method may be sensitive to this choice, yet the paper does not discuss the selection criterion or its impact.
3. The abstract claims that unreliable relation inference is a key challenge in 3D layout generation, but this central argument is not directly validated experimentally. In particular, the paper does not provide evidence showing that failures of existing methods are primarily caused by errors in relation graph inference.
4. The evaluation setup is not described clearly enough. Although the paper states that it creates test cases for 9 scene types, with 3 scenes per type and instructions at three difficulty levels, it remains unclear how these test scenes are actually obtained. In particular, the paper does not specify whether the scenes are manually designed or derived from an existing dataset, how the room sizes and asset sets are selected, or how the instructions are constructed.

---

> ### Author Rebuttal · Authors · 2026-03-31
>
> We appreciate the reviewer's thoughtful comments and address all concerns below. An anonymous PDF with figures and tables is available at: https://anonymous.4open.science/r/ICML4647-BF6F/ICML_4647_rebuttal.pdf
>
> # Q1: Consistent Intermediate States
>
> A1: Thanks for this suggestion. Here, "intermediate states" refer to the relations re-expressed after each hop. Since each hop is a relative relation defined in an object-centric frame, multi-hop reasoning requires shifting reference frames and re-expressing earlier relations under new frames. This allows small inconsistencies to accumulate across hops, leading to errors in both directional semantics and metric offsets.
>
> To support this more directly, the attached ***Table 1*** and ***Figure 1*** show that the placement error rate generally increases with the number of hops (i.e., reference-frame shifts), from **14.8%** at **Hop 1** to **47.6%** at **Hop 5+** when our modules are removed. The attached ***Figure 2*** shows a concrete reasoning trace where a reference-frame shift induces semantic drift. Together, these results provide empirical support that repeated reference-frame shifts make it harder for MLLMs to maintain consistent intermediate states. We will clarify this in the revision.
>
> # Q2: Anchor Selection
>
> A2: We would like to clarify that we do **not** impose a hand-crafted anchor-selection criterion. Instead, the MLLM determines the anchor jointly with unit construction. The only constraint is that the anchor must be the object that connects the unit to the rest of the scene. To examine the sensitivity of this design, we conduct additional repeated experiments under the same setting (attached ***Table 2***). The full model shows consistently lower run-ro-run variance than the variant without decomposition, reducing **%CR/%OR** from **0.83/1.63** to **0.31/0.90** and **Real./Func.** from **1.331/1.255** to **0.848/0.908**. This suggests that introducing decomposition and the corresponding anchor assignment improves, rather than hurts, the stability of the overall framework in practice. We will clarify this in the revision.
>
> # Q3: Unreliable Relation Inference
>
> A3: We respectfully clarify that our claim is not that unreliable relation inference is the only or primary source of failure in 3D layout generation. Rather, it is a major and under-addressed challenge that prior methods often handle through post-hoc heuristics. Our ablation in Table 3 already isolates the effect of relation inference by disabling physical loss. Under this setting, removing decomposition or imagination clearly degrades performance, indicating that improving relation inference alone already improves the final layout quality.
>
> To further examine this issue for existing methods, we reduce their downstream correction so that their final layouts more directly reflect the inferred relations. LayoutGPT directly predicts absolute poses, while Holodeck relies on coarse relations and grid-map discretization, which makes precise relational semantics harder to preserve. For LayoutVLM, we disable self-consistency decoding and physical loss. For R$^{3}$L, we likewise remove the physical loss. As shown in the attached ***Table 3*** and ***Figure 3***, without physical constraints, LayoutVLM's collision ratio increases substantially from **0.6** to **7.0**, whereas R$^{3}$L changes only slightly (from **0.0/0.0** to **1.0/1.6** in **%CR/%OR**), while maintaining nearly identical semantic scores (**7.9/7.5/9.1** vs. **8.0/7.3/9.1** for **Real./Func./Instr.**). Together, these observations support the view that unreliable relation inference is a major contributor to downstream failure, and suggest that R$^{3}$L improves over prior methods largely by making the inferred relations more reliable.
>
> # Q4: Test Cases
>
> A4: Thanks for this important point. We follow the test-case generation protocol of LayoutVLM, but do not directly reuse its benchmark because it is not fully aligned with our setting. Specifically, prior benchmarks often use short, simple instructions, which are insufficient for evaluating compositional relative spatial reasoning. In addition, our work focuses on floor objects, while prior benchmarks include many small tabletop or non-floor objects that fall outside our scope.
>
> Accordingly, our test scenes are constructed rather than directly taken from an existing dataset. For each of the 9 scene types, we construct 3 scenes with predefined room sizes, use GPT-5 to generate instructions at different difficulty levels together with plausible asset lists, retrieve 3D assets from Objaverse, and perform human verification to remove or replace low-quality assets (e.g., low-poly assets). All compared methods are evaluated on the same instructions and asset sets, ensuring a fair comparison. We will make the construction pipeline more explicit in the revision, include the generation prompt (attached ***Figure 4***), and release the benchmark upon acceptance to facilitate future research.

---

> > ### Author Rebuttal · Reviewer_VwH2 · 2026-04-03
> >
> > Thank the authors for the rebuttal. I appreciate your candid acknowledgement of the method's limitations have successfully resolved several of my initial concerns.

---

> > > ### Author Response · Authors · 2026-04-04
> > >
> > > Dear Reviewer VwH2,
> > >
> > > We sincerely appreciate your time, effort, and thoughtful feedback. Your insightful suggestions have helped us improve our manuscript.
> > >
> > > We are encouraged by your acknowledgment that our rebuttal has adequately addressed the concerns raised in your original review. We will incorporate all the discussed improvements into the camera-ready version. In light of this, we would be grateful if you could kindly consider re-evaluating the current score to better align with your post-rebuttal assessment.
> > >
> > > Thank you again for your time and consideration.
> > >
> > > Best, Authors

---

### Official Review · Reviewer_Tg6d · 2026-03-13

**Soundness:** 3
**Presentation:** 3
**Significance:** 3
**Originality:** 3
**Overall Recommendation:** 5
**Confidence:** 3

**Summary:**

The paper proposed an $R^3L$ framework to improve the reliability and consistency of 3D layout generation. The proposed framework utilizes the MLLM to decompose the object into subunits and to establish intra-unit relations based on the selected anchor unit, thereby creating local maps. Then, it creates inter-unit relationships among anchors to build the global map. This reduces the error caused by changing the perspective of the MLLM. This process is called invariant spatial decomposition. Next, the process undergoes self-refinement to ensure consistent layout locations generated by the MLLM in both local and global layouts. Lastly, the relation is expressed as differentiable constraints to identify the final location of each object based on the generated relations during the supportive spatial optimization process. The authors demonstrate the effectiveness of the proposed framework on generated test cases, following previous literature. The proposed method illustrates the improvement over several SOTA layout generation models.

**Compliance With Llm Reviewing Policy:**

Affirmed.

**Final Justification:**

The rebuttal addresses my initial weaknesses in the paper, including clarification on the use of a new benchmark, an additional baseline and results, and a hard-to-follow section.

**Key Questions For Authors:**

1. What is the performance of GPT-5 alone compared to purposed method?

2. Why are the previous benchmarks from cited papers not used in the evaluation?

3. Is there a correlation between the human evaluation and the automatic evaluation? Discussing this would further strengthen the proposed framework.

**Limitations:**

Yes

**Strengths And Weaknesses:**

# Strengths

- The paper includes a comparison with several SOTA methods using numerical analysis as well as a comprehensive human comparison between the proposed method and previous SOTA models.

- The paper provides an ablation study to ensure that each component proposed is significant and contributes to the improvement of the framework.

- The paper includes illustrations to show the improvement of the proposed method over previous SOTA models.

- The paper provides the theoretical proof that shows their method reduces the number of reference frame transformations, which could lead to observed improvement.

- The paper is well-written with a well-defined function for each proposed component.

---

# Weaknesses

- While the improvement is discussed with several metrics, the analysis of the improvement over the previous SOTA model might need to be discussed further, as well as the error analysis of the proposed method. This would help strengthen and guide future improvements in the area.

- The authors generated the evaluation test set by themselves. Even though it follows previous literature, the rationale for this decision should be discussed, explaining why the previous benchmark or data used in the cited literature cannot be used in the paper evaluation.

- If the GPT-5 was used as the baseline for the proposed framework, the comparison against this baseline might need to be discussed as well.

- While the figure is well-explained, some parts of the methodology might have some challenge to understand. Including a run-through example might make the paper easier to understand.

---

> ### Author Rebuttal · Authors · 2026-03-31
>
> We appreciate the reviewer's thoughtful comments and address all concerns below. An anonymous PDF with figures and tables is available at: https://anonymous.4open.science/r/ICML4647-BF6F/ICML_4647_rebuttal.pdf
>
> # Q1: Improvement and Error Analysis
>
> A1: Thanks for this suggestion. As shown in Table 1 and Figure 3, R$^{3}$L improves upon existing methods by addressing their different failure modes. LayoutGPT generates semantically plausible but physically invalid layouts because it directly predicts absolute object poses. Holodeck struggles to maintain semantic coherence because it relies on coarse relation modeling and grid-based solving. LayoutVLM better balances semantics and feasibility through optimization, but it remains sensitive to inaccurate or conflicting inferred relations. In contrast, R$^{3}$L reduces error accumulation in long coupled relation chains through invariant spatial decomposition, and improves global consistency through consistent spatial imagination.
>
> We also provide representative failure cases (attached ***Figure 7***) to better characterize the limitations of R$^{3}$L. Typical remaining issues include a navigability gap in some collision-free layouts, where clearance is insufficient for human passage, and errors under complex rotational configurations, where dense layouts with diverse orientations make rotation reasoning less reliable. We will discuss these failure cases in the revision.
>
> # Q2: Test Cases
>
> A2: Thanks for this important point. We follow the test-case generation protocol of LayoutVLM, but do not directly reuse its benchmark because it is not fully aligned with our setting. Specifically, prior benchmarks often use short, simple instructions, which are insufficient for evaluating compositional relative spatial reasoning. In addition, our work focuses on floor objects, while prior benchmarks include many small tabletop or non-floor objects that fall outside our scope.
>
> Therefore, we regenerate and filter the test cases to better match our setting while keeping the evaluation protocol aligned with prior work to ensure a fair comparison. Concretely, for each of the 9 scene types, we construct 3 scenes with predefined room sizes, use GPT-5 to generate instructions at different difficulty levels together with plausible asset lists, retrieve 3D assets from Objaverse, and perform human verification to remove or replace low-quality assets (e.g., low-poly or mislabeled assets). All methods are evaluated on the same instructions and asset sets. We will make the construction pipeline more explicit in the revision, include the generation prompt (attached ***Figure 4***), and release the benchmark upon acceptance to facilitate future research.
>
> # Q3: GPT-5 Baseline
>
> A3: In our experiments, the reported LayoutGPT results already serve as the GPT-5 direct-generation baseline. Concretely, this baseline uses GPT-5 to directly predict absolute object poses, without our additional decomposition, imagination, or optimization modules. Compared with this direct GPT-5 baseline, R$^{3}$L improves the average **%CR/%OR** from **4.3/8.1** to **0.0/0.0**, and improves **Real./Func./Instr.** from **5.9/5.5/8.1** to **7.9/7.5/9.1** (Table 1). The human study shows the same trend, improving **Real./Func./Instr.** from **4.1/4.1/5.7** to **6.7/6.5/7.9** (Table 4). We will clarify this to avoid confusion.
>
> # Q4: Run-through Example
>
> A4: Thanks for this helpful suggestion. We provide a step-by-step example in the attached ***Figure 8***. It illustrates how R$^{3}$L performs invariant spatial decomposition, consistent spatial imagination, and supportive spatial optimization. We will incorporate this run-through example in the revision to improve the clarity of our method.
>
> # Q5: Human-LLM Alignment
>
> A5: Thank you for this suggestion. We compute Kendall’s Tau between the rankings induced by human judgments and automatic scores in the attached ***Table 8***. We observe consistent positive alignment between human evaluation and the LLM-based evaluator across all three semantic metrics: **0.52** for realism, **0.50** for functionality, and **0.56** for instruction following. Importantly, these values are close to the User-User agreement levels (**0.56/0.55/0.53**), suggesting that the LLM-based evaluation is broadly consistent with human judgment. We will include this discussion in the revision to further justify the use of LLM-based automatic evaluation.

---

> > ### Author Rebuttal · Reviewer_Tg6d · 2026-04-04
> >
> > Thank you, authors, for the detailed rebuttal addressing my concerns. I would increase my score, leading to the acceptance of the paper.

---

> > > ### Author Response · Authors · 2026-04-04
> > >
> > > Dear Reviewer Tg6d,
> > >
> > > We sincerely appreciate your time, effort, and thoughtful feedback. Your insightful suggestions have helped us improve our manuscript.
> > >
> > > We are very grateful for your positive follow-up and updated evaluation. We are encouraged that our rebuttal has adequately addressed your concerns.
> > >
> > > Best, Authors

---

### Decision · Program_Chairs · 2026-04-30

**Decision:**

Accept (regular)

**Comment:**

This paper introduces a framework for instruction-based 3D layout generation to improve relative spatial reasoning, including decomposition, imagine-and-revise, and spatial optimization. The reviews are overall positive: 1 weak rejection (but said their concerns are resolved) and 3 positive ratings. The main concerns are that (1) the evaluation setup, including the new benchmark, was not initially clear, (2) some key method/workflow details were under-explained, and (3) the paper’s central claim about repeated reference-frame shifts / relation inference as a major source of failure was not fully validated. In the rebuttal, the authors clarified the benchmark construction and scope, clarified that LayoutGPT already serves as the direct GPT-5 baseline, added a run-through example, and further analyzed anchor selection and hop-wise errors. Overall, it addressed most reviewers' concerns while substantially improving the paper's clarity.

Given the strong results and positive post-rebuttal feedback, I recommend Accept. I also note that the lone reject score appears inconsistent with the reviewer’s post-rebuttal acknowledgement, which states that the concerns were fully resolved.